# Re-ttention: Ultra Sparse Visual Generation via Attention Statistical Reshape

**Ruichen Chen**
ECE Department
University of Alberta
ruichen1@ualberta.ca

**Keith G. Mills**
Division of CSE
Louisiana State University
keith.mills@lsu.edu

**Liyao Jiang**
ECE Department
University of Alberta
liyao1@ualberta.ca

**Chao Gao**
Huawei Technologies
Edmonton, Alberta, Canada
chao.gao4@huawei.com

**Di Niu**
ECE Department
University of Alberta
dniu@ualberta.ca

## Abstract

Diffusion Transformers (DiT) have become the de-facto model for generating high-quality visual content like videos and images. A huge bottleneck is the attention mechanism where complexity scales quadratically with resolution and video length. One logical way to lessen this burden is sparse attention, where only a subset of tokens or patches are included in the calculation. However, existing techniques fail to preserve visual quality at extremely high sparsity levels and might even incur non-negligible compute overheads. To address this concern, we propose Re-ttention, which implements very high sparse attention for visual generation models by leveraging the temporal redundancy of Diffusion Models to overcome the probabilistic normalization shift within the attention mechanism. Specifically, Re-ttention reshapes attention scores based on the prior softmax distribution history in order to preserve the visual quality of the full quadratic attention at very high sparsity levels. Experimental results on T2V/T2I models such as CogVideoX and the PixArt DiTs demonstrate that Re-ttention requires as few as 3.1% of the tokens during inference, outperforming contemporary methods like FastDiTAttn, Sparse VideoGen and MInference.

## 1 Introduction

Diffusion Transformers (DiT) [33, 3, 2, 23, 10] combine the attention [38] mechanism with the iterative denoising of Diffusion Models [35] to generate high-quality visual content such as videos [47, 54] and images [20, 44, 41, 31]. However, a key bottleneck to generating longer videos and higher resolution content is the global properties of the self-attention module, whose compute cost scales quadratically with sequence size, i.e., resolution and video length.

Sparse attention techniques [6] aim to lower the computational burden by reducing the number of sequence tokens/patches that the attention mechanism considers during inference. Contemporary techniques like MInference [18] and Sparge Attention [51], as well as XAttention [46] achieve ~50% sparsity (i.e., reducing only 50% of the attention computations) by relying on downsampling the attention map or anti-diagonal scoring, respectively. In parallel, several recent methods [42, 48, 50] have been proposed specifically for DiTs, increasing the attention sparsity to ~70% by exploiting the inherent characteristics of diffusion process such as the progressively denoising structure and the spatial/temporal locality of attention.

39th Conference on Neural Information Processing Systems (NeurIPS 2025).

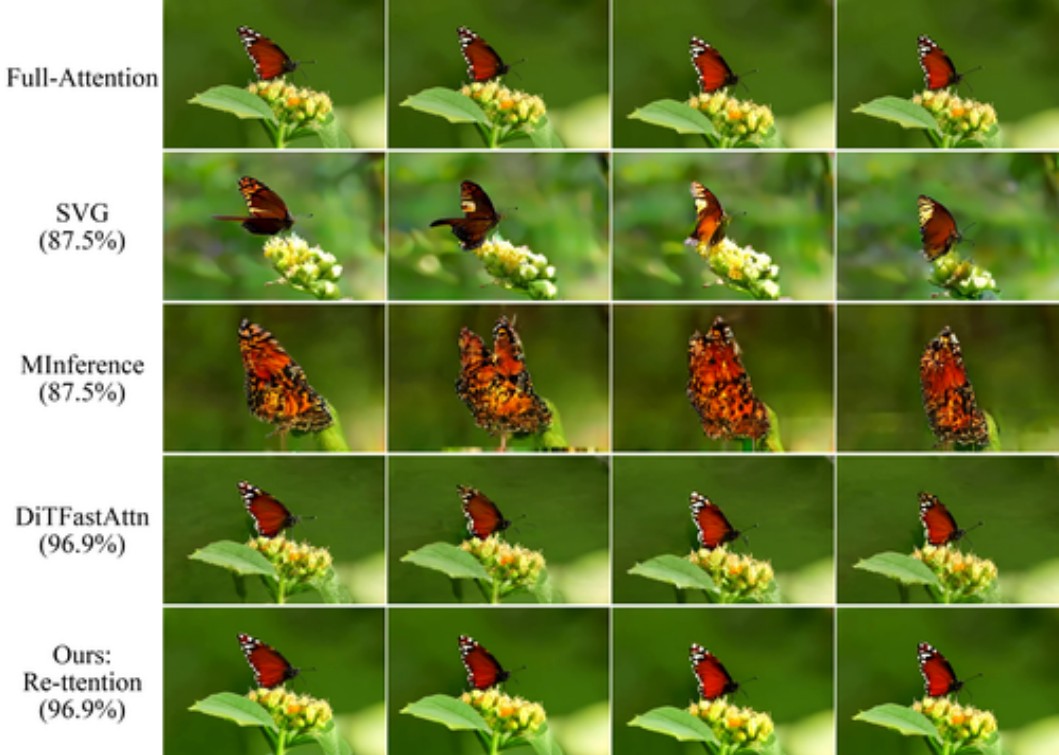

Figure 1: Visual comparison using CogVideoX-2B [47] T2V model. Columns correspond to different frames. Rows correspond to to different sparse attention methods (sparsity degree in paranthesis; higher is better). Prompt: "a colorful butterfly perching on a bud". More examples in the Appendix.

While these methods can reduce more than half of the attention computation, their effectiveness remains limited for the growing computational demands of high-resolution image and video generation. Previous researches like LongFormer [1] and BigBird [49] can achieve >95% sparse attention. However, their reliance on retraining and fine-tuning introduces significant computation burdens, limiting their applicability to modern large-scale, pretrained generative models. Thus, the development of sparse attention techniques that achieve >95% sparsity with minimal visual quality loss remains an open challenge.

In this paper, we propose an effective method to statistically reshape the attention distribution distorted by the deployment of sparse attention, which we call Re-ttention. Re-ttention overcomes the high sparsity challenge faced by the training-free sparse attention method. Moreover, it is simple to implement and incurs negligible overhead compared to standard sparse attention at the same sparsity level. Figure 1 provides sample content from our technique compared to other sparse attention methods. Our detailed contributions are as follows:

1. We relate the failure to achieve degradation-free >95% sparsity without training from scratch to the distributional shift in attention scores caused by the reduced softmax denominator term, i.e., the row-wise sum of the exponentials of involved elements. We design an experiment to illustrate the importance of preserving this term and the impact on visual generation.

2. We discover the softmax distribution redundancy among neighboring denoising steps. Although the actual value of denominator changes unpredictably, the ratio between the sparse and full denominator is relatively stable.

3. We propose that the attention scores shifted by sparse attention are viable to be recovered by approximating the *real* softmax denominator.

4. The recovered attention scores deviate from a valid probability distribution, as their sum is less than one, violating the normalization property of softmax. We leverage the redundancy among neighboring denoising steps to compensate the missing probability with residual.

We apply Re-ttention to T2V models such as CogVideoX [47] in order to outperform contemporary methods like FastDiTAttn [48], Sparse VideoGen [42] and MInference [18] on relevant tasks such as VBench [16]. Furthermore, we apply Re-ttention to T2I models like the PixArt series [3, 2] and others [23] to maximize performance on Human Preference Score v2 (HPSv2) [41] and GenEval [12] while achieving a high sparsity of 96.9%.

## 2   Related Work

**Diffusion Models** (DM) [14] dominate visual generation tasks. Early DMs [35, 34] use convolutional U-Net structures [36] as their backbones. Later, Diffusion Transformers (DiT) [33, 5] adopt the attention-based [38] of Vision Transformers (ViT) [9] to increase scalability and visual generation quality. In addition to being the favored backbone structure for text-to-image (T2I) DMs [3, 2, 23, 10, 20, 44], the DiT structure is extensible to video generation [40, 19] as well. Specifically, Latte [30] proposes a 2D+1D attention block for video generation, which performs spatial and temporal attention separately. Subsequent works like CogVideoX [47] and OpenSora [24] adopt a 3D attention structure which processes the spatial and temporal dimensions simultaneously, yielding improved generation quality. However, this enhancement comes at the cost of significantly increased computation due to the quadratic complexity of attention, highlighting the pressing need for more efficient and sparse attention, which we explore in this work.

**Sparse Attention** denotes a class of techniques that aim to alleviate the hardware cost of the attention mechanism by omitting computation for unnecessary query-key pairs. Specifically, it is well documented that the attention mechanism produces sparse results [8, 28], yet suffers from a burdensome quadratic complexity and wasted computation by default. LongFormer [1] proposes sliding window attention that restricts attention to a local region. BigBird [49] and Mistal-7B [17] extend this idea to fine-grained attention masks, while SwinFormer [29] use local attention for efficient ViT design. Although these methods can reduce the attention computation by a factor or $8\times$ or more, they often necessitate training or fine-tuning the model, thus restricting the scope of deployment.

There are also training-free sparse attention methods. MInference [18] downsamples the attention probability matrix ($QK^T$) into blocks then dynamically select the top-$k$ blocks to perform sparse attention. Subsequent research like FlexPrefill [21], Sparge Attention [51] and XAttention [46] rely on the block selection idea and propose dynamic block sorting algorithms. Further methods like StreamingLLM [43], DiTFastAttn [48] and Sparse VideoGen [42] identify the special attention patterns in LLM and DiT and propose efficient attention masking based on those patterns. However, the sparsity achievable by these methods is limited to $< 70\%$, which is much higher compared to prior works that require re-training or fine-tuning. We aim to address this gap and provide a training-free sparse attention method that can achieve $> 95\%$ sparsity on visual generation tasks.

**Caching** is a technique used in computer systems to temporarily store data or computations, thereby reducing redundant processing and improving overall efficiency. In DiT, the lengthy denoising process makes it well-suited for the application of caching techniques. Recent methods [4, 27, 53] re-use the attention outputs or the intermediate features at different denoising timesteps to skip the attention computation. Methods like DiTFastAttn [48, 50] leverage the caching mechanism to improve the visual quality.

## 3   Background: The Attention Mechanism and Sparsity

The attention mechanism [38] is the foundation of transformer architectures like DiTs. Let $X \in \mathbb{R}^{T \times d}$ be an input token/patch sequence, where $T$ is the sequence length, dependent on the input size (e.g., image resolution or video length) and $d$ is the embedding dimension, a hyperparameter of the transformer model. The attention mechanism contains $h$ heads such that $d_h = \frac{d}{h}$; $d_h \in \mathbb{Z}^+$. We first map $X$ into three representations, Query ($Q$), Key ($K$) and Value ($V$) of identical size $\mathbb{R}^{h \times T \times d_h}$, then compute the attention as follows,

$$\text{Attention}(Q, K, V) = \text{Softmax}(\frac{QK^T}{\sqrt{d_h}})V. \tag{1}$$

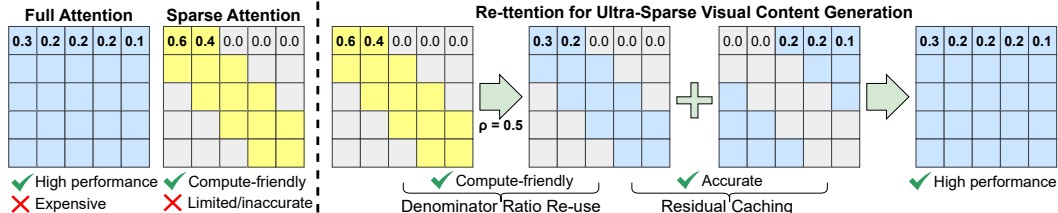

Figure 2: Illusion of attention map $A$ computed by full attention, contemporary sparse attention (window-based) and our proposed Re-ttention. Sparse attention shifts the distribution of attention scores, resulting in degraded performance as sparsity increases. In contrast, Re-ttention re-uses the denominator ratio cached from the previous denoising steps to scale the sparse attention score to the full attention level. Then, we apply residual caching to accurately restore the full attention scores.

We can decompose this mechanism into several intermittent matrices, specifically the product of $Q$ and $K$ before ($A^{\text{pre}}$) and after ($A$) the softmax operation:

$$A^{\text{pre}} = \frac{QK^T}{\sqrt{d_h}} \in \mathbb{R}^{h \times T \times T}, \qquad (2) \qquad A = \text{Softmax}(A^{\text{pre}}) \in [0,1]^{h \times T \times T}. \qquad (3)$$

The computation of these matrices is very expensive [11]. To make matter worse, their size scales quadratically with $T$, which depends on the image/video resolution and video length. However, the softmax operation is computed row-wise and produces a probability distribution $\sum_{j=1}^{T} A_{:,:,j} = 1$, which empirically produces a sparse $A$ in practice [8, 28].

Therefore, one way to alleviate this computational burden is to use a sparse attention mechanism. The key idea is to *omit* less relevant values of $A^{\text{pre}}$, that are likely to be 0 or close to 0 in $A$, from the softmax computation altogether. Formally, we express the sparse attention calculation using a mask $M \in \{0,1\}^{h \times T \times T}$ where 1 means an index of $A^{\text{pre}}$ will be included in the softmax, while the rest are excluded. The indexes of the included values in $A^{\text{pre}}$ form a set $\mathcal{S} = \bigcup_{k=1}^{h} \bigcup_{i=1}^{T} \mathcal{S}_{k,i}$, where $\mathcal{S}_{k,i} = \{(k,i,j) | M_{k,i,j} = 1, 0 \leq j \leq T\}$.

Given an arbitrary element of the pre-softmax matrix $A^{\text{pre}}_{k,i,j}$, the normal and sparse softmax computation are given by

$$A_{k,i,j} = \frac{\exp(A^{\text{pre}}_{k,i,j})}{\sum_{t=1}^{T} \exp(A^{\text{pre}}_{k,i,t})}, \quad (4) \qquad A_{k,i,j} = \begin{cases} \dfrac{\exp(A^{\text{pre}}_{k,i,j})}{\sum_{t \in \mathcal{S}_{k,i}} \exp(A^{\text{pre}}_{k,i,t})} & \text{if } j \in \mathcal{S}_{k,i}, \\ 0 & \text{otherwise,} \end{cases} \qquad (5)$$

respectively. Ultimately, the mask matrix $M$ determines the potential amount of computational savings. $M$ can be computed statically [7] prior to inference or dynamically [18, 21, 51, 46] at runtime. Static techniques make more assumptions about the sparse regions of $A$ while dynamic techniques impose additional inference overhead to compute $M$.

Regardless of technique, we can quantify the attention sparsity as a percentage, e.g., 10%, 50%, 90%, etc., simply by computing the ratio of values in $M$ that are 0 as follows:

$$\text{Sparsity} = \left(1 - \frac{|\mathcal{S}|}{hT^2}\right) \times 100\%, \qquad (6)$$

where a higher value for sparsity corresponds to a lower computational burden. Therefore, sparse $M$ corresponds to an overall sparse attention. However, high sparsification can cause significant shifts in the softmax calculation statistics [43] and lead to detrimental performance. As we will next show, our proposed method, Re-ttention, aims to identify these statistical issues and address them.

## 4 Proposed Method: Re-ttention

In this section we form a hypothesis regarding how distributional shift in softmax statistics prevents current training-free sparse attention methods from satisfactorily operating at high sparsity, e.g.,

$> 95\%$. We then elaborate on our proposed Re-ttention technique, which overcomes this burden by re-using and caching softmax statistics at high sparsity. Figure 2 provides a high-level overview of our proposed technique in comparison to full and sparse attention.

## 4.1 Importance of the Softmax Denominator

As a preliminary investigation, we gauge the performance of several existing sparse attention techniques [18, 48, 42]. We consider the GenEval [12] benchmark and evaluate performance across a spectrum of sparsity values, i.e., starting at the highest sparsity these techniques consider in their original manuscript and then further increasing the sparsity.

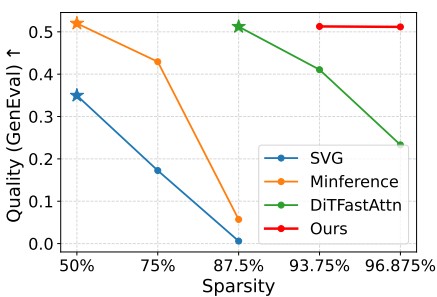

Figure 3illustrates our findings. We observe that existing approaches suffer a monotonic performance drop when the sparsity is further increased beyond their proposed value (denoted with ★). Per Eq. 5, a higher value of sparsity corresponds the inclusion of fewer tokens in the softmax denominator as $\mathcal{S}_{k,i}$ shrinks. Thus, the further removal of tokens, i.e, increasing sparsity closer to 100%, has a larger impact on the overall denominator value [43]. This phenomenon, introduces a detrimental distribution shift in the overall attention scores.

Figure 3: Quality-sparsity comparison of Re-ttention, Sparse VideoGen (SVG), MInference and DiTFastAttn. ★ denotes the sparsity level that prior methods operate under non-degraded conditions.

We design a toy experiment to test this hypothesis and showcase the significance of the softmax denominator term. Specifically, we define a post-softmax masking operation as

$$A' = A \circ M, \tag{7}$$

where $A$ is the output of the original full softmax attention via Eq. 4, $\circ$ denotes element-wise multiplication and $A'$ is masked attention. We emphasize that Equation 7 is not a proper sparse attention calculation and does not entail speedup. *However*, it *mimics* the output of sparse attention as $M$ still zeroes out the same indices of $A$, yet preserves the denominator of the full softmax.

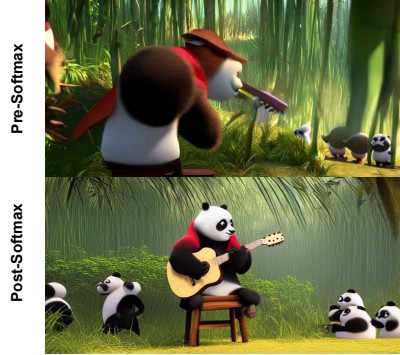

We calculate $M$ using sliding window attention [1]. We then generate visual content using both the formal sparse attention from Equation 5 and our post-softmax Equation 7 for comparison. Figure 4 provides a comparison, though we provide additional examples and prompts in the supplementary due to space constraints. We observe how the post-softmax attention preserves the guitar-playing panda, chair and background while the pre-softmax attention creates a noisy frame with jumbled contents where the panda appears to eat the guitar. Thus, these visual results validate our assumption regarding the importance of maintaining the softmax denominator. The challenge now becomes how to preserve this information in an efficient sparse attention setup.

Figure 4: Visual comparison of pre-Softmax and post-Softmax masking on CogVideoX-2B with 66% sparsity, using sliding-window attention [1].

## 4.2 Leveraging Denoising Properties for Statistical Reshape

One way to mitigate the distribution shift is to maintain the softmax denominator from the full attention calculation. We achieve this by exploiting the sequential nature of the DM denoising process and taking inspiration from DiT caching [4, 27, 53] and redundancy [37] methods.

**Denominator Approximation.** Figure 5 tracks the magnitude of the softmax denominator for a single token in the 9th head of the 12th DiT block in PixArt-$\alpha$ Specifically, we calculate the

denominator using both the full and sparse attention (with 87.5% sparsity) as well as the ratio $\rho$ between these statistics,

$$\rho = \frac{\sum_{t \in \mathcal{S}_{k,i}} \exp(A^{\text{pre}}_{k,i,t})}{\sum_{t=1}^{T} \exp(A^{\text{pre}}_{k,i,t})}. \qquad (8)$$

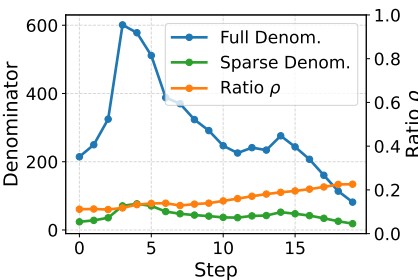

Figure 5: Plotting softmax denominators for full and sparse attention as well as the ratio $\rho$ per Eq. 8 across 20 steps.

This yields an insightful observation: While the actual value of the denominators may change unpredictably and non-monotonically over the denoising process, given a statically computed mask $M$, the ratio $\rho$ follows a predictable trend. Therefore, we propose a simple method to recover the attention distribution by modifying the output of the sparse softmax operation Eq. 5 as follows:

$$A'_{k,i,j} = \rho A_{k,i,j}, \qquad (9)$$

where we cache $0 < \rho \leq 1$ from a previous step. Multiplying the Softmax output of sparse attention by $\rho$ approximates the full attention value, mitigating the distribution shift. In practice, since $\rho$ is not a constant per Fig. 5, we empirically find that slightly increasing the cached $\rho$ after each denoising step achieves better performance. Thus, we parameterize $\rho_t = \rho_{t-1} + \lambda$ after each denoising step, where $\lambda$ is a ramp-up hyperparameter.

**Residual Caching.** Although we can modify the softmax to make the sparse attention more closely match that of full attention, Equation 8 does not yield proper probability distributions as $\sum_{j=1}^{T} A_{:,:,j} < 1$. Practically, this reduces the magnitude of overall attention outputs per Equation 1, which can negatively impact performance. To address this issue, we first define the residual $R$ as the difference between the full attention computed via Equation 4 and our $\rho$-reshaped attention via Equations 5 and 9:

$$R = \text{FullAttention}(Q, K, V) - \text{ReshapeAttention}(Q, K, V, \rho). \qquad (10)$$

We can later add $R$ to our sparse attention output. In fact, $R$ is mathematically equivalent to the attention output of the masked tokens at the caching timestep.

Overall, the idea behind Re-ttention is to compute sparse attention for important regions, while re-using previously cached statistics from previous steps in less important regions of the attention map. This involves caching necessary statistics from the full attention in some steps, which is common in DiT sparse attention methods [48, 50].

## 5 Experimental Setup and Results

We evaluate Re-ttention on both the text-to-video (T2V) and text-to-image (T2I) tasks using a number of DiT models, such as CogVideoX (2B) [47], PixArt-$\alpha$/$\Sigma$ (0.6B) [3, 2] and Hunyuan-DiT (1.6B) [23]. We generate $720 \times 480$ resolution, 6 second videos at 8 fps and $1024 \times 1024$ pixel images throughout this paper. We compare to several existing sparse attention methods for visual content in the literature like Sparse VideoGen (SVG) [42], MInference [18] and DiTFastAttn [48, 50] to demonstrate both qualitative and quantitative performance gains and computational cost savings.

**Implementation Details.** Specifically, we use the HuggingFace Diffusers library [39] to instantiate the base DiT models and consider the default values for inference parameters like the classifier-free guidance (CFG) scale and number of denoising steps - 50 for CogVideoX/Hunyuan and 20 for the PixArt DiTs. Following prior literature on DiT acceleration [42, 53, 22, 27], we apply the full attention during the first 5, 10 or 15 steps for the PixArt DiTs, Hunyuan and CogVideoX models, respectively, and then apply the sparse attention mechanism for the remainder of the denoising process. Further, we set a caching period of 5 steps for DiTFastAttn and Re-ttenion, where we

Table 1: Quantitative evaluation results for T2V model CogVideoX-2B [47] on VBench [16] and other metrics. Arrows indicate if a higher or lower value of a metric is preferred. Best and second-best results in **bold** and *italics*, respectively.

| Attention | Sparsity ↑ | PSNR ↑ | SSIM↑ | LPIPS ↓ | ImageQual ↑ | SubConsist ↑ |
|---|---|---|---|---|---|---|
| Full-Attention | 0.0% | Reference | Reference | Reference | 65.72% | 94.97% |
| SVG | 87.5% | 14.48 | 0.548 | 0.501 | 54.48% | 89.26% |
| SVG | 96.9% | 10.50 | 0.418 | 0.898 | 51.82% | **96.73%** |
| MInference | 87.5% | 14.99 | 0.558 | 0.480 | 53.78% | 83.71% |
| MInference | 96.9% | 9.25 | 0.325 | 0.818 | 34.36% | 75.84% |
| DiTFastAttn | 96.9% | *27.93* | *0.865* | *0.098* | *64.86%* | 94.32% |
| Re-ttention | 96.9% | **27.96** | **0.894** | **0.059** | **64.87%** | *94.80%* |

perform full attention to cache required statistics. For fair comparison, we apply this caching to SVG and MInference as well: SVG and MInference will perform full attention at the same timesteps as DiTFastAttn and Re-ttention. To perform T2I using SVG, we treat the image as a video containing a single frame. We provide further baseline experimental details in the supplementary.

The rest of this section is organized as follows: We enumerate our T2V and T2I evaluation setup and results in Sections 5.1 and 5.2, respectively. Next, we provide ablation studies in Section 5.3.

## 5.1  Text-to-Video Evaluation

We perform quantitative T2V evaluation using the Animal and Architecture categories of VBench [16], which consist of 100 videos each. For video quality, we use VBench score to evaluate standalone video quality. Specifically, we follow previous literature [42] and report the Image Quality and Subject Consistency metrics in VBench. Additionally, we compute the Peak Signal-to-Noise Ratio (PSNR) [15], Structural Similarity Index Measure (SSIM) [32] and Learned Perceptual Image Patch Simularity (LPIPS) [52]. These metrics evaluate the similarity and quality of videos generated by sparse attention methods relative to those generated using the full attention mechanism. We evaluate all methods at 96.9% sparsity to provide an apples-to-apples performance investigation. However, some methods exhibit substantial degradation at this level and produce very noisy/black frames, so we additionally report results at a less aggressive setting of 87.5% sparsity.

Table 1 presents our findings. Results demonstrate that Re-ttention consistently outperforms all other baselines in terms of video quality and similarity metrics. Notably, Re-ttention not only outperforms both SVG and MInference at the strict sparsity of 96.9%, but also at 87.5% sparsity. The one exception is SVG at 96.9% sparsity, which achieves the highest SubConsist performance. However, this result is an outlier, as it even exceeds the SubConsist performance of full attention significantly while SVG substantially underperforms on all other metrics at this sparsity level. Furthermore, Re-ttention also outperforms DiTFastAttn, which also involves caching additional statistics at the high sparsity level of 96.9%. Therefore, overall, these results demonstrate the robustness, competitive performance of Re-ttention at $> 95\%$ sparsity in T2V applications.

We provide some sample frames from videos generated by Re-ttention, baseline sparse attention methods and full attention. Specifically, recall Figure 1 in the introduction. The video generated by Re-ttention shows the best clarity and temporal consistency across frames for the main subject, and it has no artifacts in the background. Moreover, the video generated by Re-ttention is most similar to the reference video generated with full-attention. In contrast, the video generated by DiTFastAttn has noisy texture artifacts both in the background and the subject. For SVG and MInference, the subject is inconsistent and deformed despite using a much lower sparsity. We provide more T2V visual comparisons in the supplementary materials.

## 5.2  Text-to-Image Results

We evaluate T2I performance on a comprehensive set benchmark metrics: GenEval [12], HPSv2 [41], and MS-COCO 2014 [25]. GenEval consists of 553 unique prompts. For each prompt, the DiT generates 4 images. HPSv2 consists of four image categories: Animation, Concept-art, Painting and Photos. Each category consists of 800 images for 3.2k generations in total. Finally, we generate 10k

Table 2: Quantitative evaluation results for PixArt-$\alpha$ [3], PixArt-$\Sigma$ [2] and Hunyuan-DiT [23] across the GenEval [12], HPSv2 [41], and MS-COCO 2014 [25] benchmarks. Best and second best results in **bold** and *italics*, respectively.

| Model | Attention | Sparsity ↑ | GenEval ↑ | HPSv2↑ | LPIPS ↓ | IR ↑ | CLIP ↑ |
|---|---|---|---|---|---|---|---|
| PixArt-$\alpha$ | Full-Attention | 0.0% | 0.480 | 30.79 | Reference | 0.864 | 31.28 |
| | SVG | 75.0% | 0.368 | 25.24 | 0.655 | -0.141 | 29.43 |
| | MInference | 75.0% | 0.433 | *28.04* | 0.458 | 0.549 | 30.93 |
| | DiTFastAttn | 93.8% | 0.431 | 27.26 | 0.506 | **0.688** | 30.72 |
| | DiTFastAttn | 96.9% | 0.364 | 26.71 | 0.590 | 0.314 | 29.63 |
| | Re-ttention | 93.8% | **0.456** | **28.29** | **0.354** | **0.688** | **31.21** |
| | Re-ttention | 96.9% | *0.448* | 27.57 | *0.372* | *0.646* | *31.20* |
| PixArt-$\Sigma$ | Full-Attention | 0.0% | 0.544 | 30.70 | Reference | 0.953 | 31.54 |
| | SVG | 75.0% | 0.172 | 18.48 | 0.742 | -1.315 | 26.09 |
| | MInference | 75.0% | 0.429 | 27.09 | 0.536 | 0.457 | 30.76 |
| | DiTFastAttn | 93.8% | 0.411 | 27.64 | 0.591 | 0.507 | 30.08 |
| | DiTFastAttn | 96.9% | 0.233 | 22.79 | 0.734 | -0.600 | 27.37 |
| | Re-ttention | 93.8% | **0.513** | **28.37** | **0.417** | **0.808** | **31.59** |
| | Re-ttention | 96.9% | *0.512* | *27.72* | *0.435* | *0.784* | **31.59** |
| Hunyuan | Full-Attention | 0.0% | 0.610 | 30.41 | Reference | 1.027 | 31.77 |
| | SVG | 75.0% | 0.317 | 24.73 | 0.854 | -0.574 | 27.92 |
| | MInference | 75.0% | 0.450 | 23.94 | 0.720 | -0.063 | 30.15 |
| | DiTFastAttn | 93.8% | 0.024 | 14.77 | 0.896 | -2.074 | 22.47 |
| | DiTFastAttn | 96.9% | 0.002 | 12.28 | 0.923 | -2.237 | 22.10 |
| | Re-ttention | 93.8% | *0.585* | **29.03** | **0.598** | *0.911* | *31.63* |
| | Re-ttention | 96.9% | **0.590** | *28.89* | *0.606* | **0.923** | **31.65** |

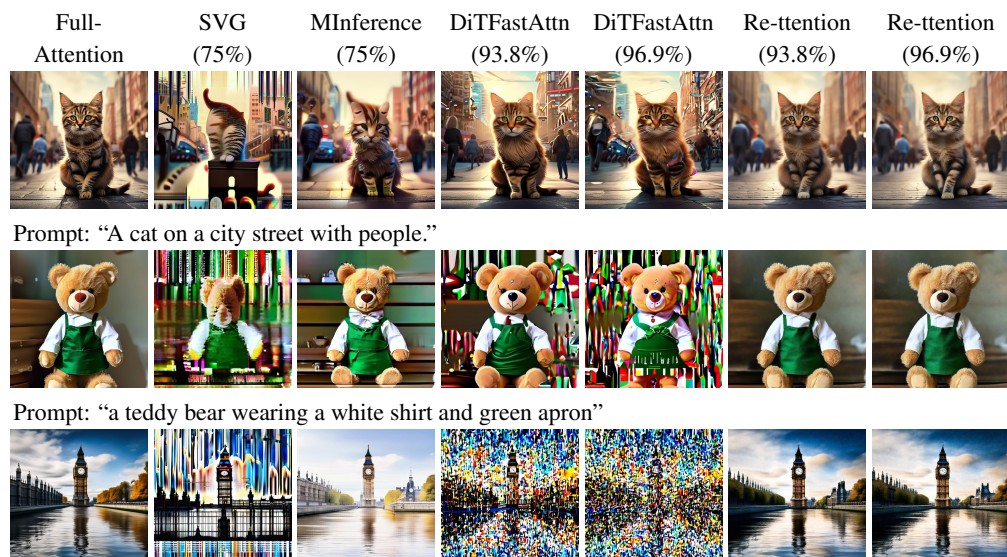

| Full-Attention | SVG (75%) | MInference (75%) | DiTFastAttn (93.8%) | DiTFastAttn (96.9%) | Re-ttention (93.8%) | Re-ttention (96.9%) |

Prompt: "A cat on a city street with people."

Prompt: "a teddy bear wearing a white shirt and green apron"

Prompt: "A view of Big Ben from over the water, during the day."

Figure 6: Visual comparison on MS-COCO 2014 [25] prompts using PixArt-$\alpha$ (row 1), PixArt-$\Sigma$ (row 2), and Hunyuan (row 3). We show images generated by Re-ttention (our method) and by other attention methods in different columns. We provide further examples in the appendix.

images using the MS-COCO 2014 validation set and measure the LPIPS score [52], ImageReward (IR) [45] and CLIP score [13] using the ViT-B/16 backbone.

Table 2 lists our results on the T2I task. Re-ttention outperforms all other sparse attention methods across models and metrics, showing consistently better performance. Additionally, Re-ttention achieves this while operating under an extremely high sparsity of 96.9%, which reduces the token/patch sequence to less than one *twentieth* of its original size, whereas other baseline methods underperform at 75% sparsity, which only reduces sequence length down to one *fourth*. Additionally, our performance on the IR metric is consistently positive, a feat that no other sparse attention method attains. Moreover, while DiTFastAttn attains similar T2V performance to Re-ttention, it fails to generalize to the T2I task on Hunyuan in terms of GenEval and HPSv2 performance, even at reduced sparsity of 93.8%. In contrast, Re-ttention performance neither suffers at 93.8% nor 96.9% sparsity, underscoring the effectiveness of our technique.

Next, we present the visual (qualitative) comparisons on PixArt-$\alpha$ [3], PixArt-$\Sigma$ [2] and Hunyuan [23] T2I models in Figures 6. More visual comparisons can be found in the Appendix. Overall, Re-ttention generates images with better image quality than other sparse attention methods and has higher similarity to the reference images generated by full-attention, even when using an extreme sparsity of 96.9%. For PixArt-$\alpha$ [3] and PixArt-$\Sigma$ [2], Re-ttention generates clean, high-quality images that are well aligned to the prompts. Whereas the other methods often generate colored noise artifacts, distorted subjects, and lower quality images. For Hunyuan [23], we observe that the other sparse attention methods generate severely degraded images, while Re-ttention can generate images that are similar to images generated by full-attention.

## 5.3  Ablation Studies

Finally, we ablate the effect of the ramp-up hyperparameter $\lambda$ on PixArt-$\Sigma$ [2] on overall performance. Specifically, we evaluate on HPSv2 overall as well as the 'animation' and 'concept-art' categories. Table 3 reports our findings. These findings demonstrate the robustness of Re-ttention as it is possible to forgo the $\lambda$ parameter, yet it is better to select a moderate value.

Table 3: HPVs2 score under different ramp-up hyperparameter $\lambda$ with 96.9% sparsity on PixArt-$\Sigma$ [2].

| $\lambda$ | Anime ↑ | ConceptArt ↑ | HPSv2 ↑ |
|---|---|---|---|
| 0 | 29.40 | 26.94 | 27.46 |
| 0.01 | 28.89 | 26.21 | 26.83 |
| 0.02 | 28.88 | 26.19 | 26.82 |
| 0.04 | **29.60** | **27.23** | **27.72** |

## 6  Conclusions and Future Work

We propose Re-ttention, a training-free sparse attention method for Diffusion Transformers, which achieves 96.9% sparsity without performance loss on DiTs like CogVideoX and Hunyuan. We attain these gains by identifying the distribution shift of attention scores incurred by sparse attention methods that prevents extreme sparsity ($> 95\%$) without significant performance degradation and resolve this issue using a combination of caching and statistical re-use. We evaluate Re-ttention on T2V and T2I tasks, outperforming contemporary baselines like SVG, MInference and DiTFastAttn.

Potential future directions to expand Re-ttention and address limitations should aim to repurpose our contributions in the application domain of LLMs or autoregressive visual content generation models. These models rely on causally masked attention, meaning that our attention statistical reshape, which leverages the step-wise denoising process in diffusion models to reuse cached attention statistics from previous steps, must be handled differently in this setting where such sequential caching is unavailable. Also, Re-ttention implements sparse attention using a static mask, though further investigation is merited to validate it in the context of dynamically-generated sparse attention masks. Dynamically adapting the sparsity pattern based on attention statistics or token importance could further improve efficiency while preserving output quality, enabling Re-ttention to generalize across a wider range of sequence modeling and generative tasks.

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

# A  Appendix

We provide additional information about our work. Sections A.1 and A.2 provide statements about the broader impacts of our work and limitations, respectively. Further, we provide additional details on our methodology in Sec. A.3 and baselines in Sec. A.4. Section A.5 provides elaborates on our pre vs. post-softmax example from Figure 4. Finally, Sections A.6 and A.7 provide additional T2V and T2I results, respectively.

## A.1  Societal Impacts

Re-ttention improves the efficiency of image and video generation by enabling extremely sparse attention. This makes high-quality generative models more accessible and environmentally sustainable by reducing computational and energy demands. By lowering resource barriers, Re-ttention can benefit creators, educators, and researchers in low-resource settings. While any generative model carries a risk of misuse, Re-ttention does not introduce new risks beyond existing systems. Responsible deployment and continued dialogue on ethical use remain important.

## A.2  Limitation

We design Re-ttention around achieving high sparsity for the non-autoregressive self-attention mechanism utilized by visual generation DiTs, rather than the autoregressive, causally-masked attention of LLMs which may more often feature different attention patterns such as columns [7, 43]. Additionally, Re-ttention exploits the sequential nature of DMs and is inspired by DiT caching techniques [4, 27, 53]. Our method may not be readily generalizable to autoregressive LLMs, though modifications and expansions into this field are a potential future work. Furthermore, Re-ttention is currently designed for statically computed attention masks, which offer speedup advantages. Extending the approach to support dynamically computed masks to support fine-grained sparse attention presents a promising direction for future work.

Although we did not implement a custom GPU kernel, we measured inference latency on typical GPUs and observed that Re-tention achieves comparable runtime to DiTFastAttn across all tested models. This demonstrates that our contributions do not impose significant computational overhead, confirming that Re-tention maintains both high sparsity and practical efficiency.

## A.3  Explanation of Re-ttention

In Section 4 we claim that the residual $R$ in Re-ttention is mathematically equivalent to the attention output of the masked tokens at the caching timestep. We now further elaborate on this claim:

Recall the definition of $A$ in Eq. 3 and the set $S$ that contains the included values (by sparse attention) in $A$. Hence, the $A$ can be decomposed into two parts:

$$
\begin{aligned}
A &= A_{\in S} + A_{\notin S}, \\
A_{\in S} &= A \circ \mathbf{1}_{(k,i,j) \in S}, \\
A_{\notin S} &= A \circ \mathbf{1}_{(k,i,j) \notin S},
\end{aligned}
\tag{11}
$$

where $\mathbf{1}_{\in S}$ is the indicator matrix that is 1 where $(k, i, j) \in S$, and 0 elsewhere. Conversely, $\mathbf{1}_{\notin S}$ is 1 where $(k, i, j) \notin S$ and 0 elsewhere.

At the caching timestep, we have the ratio $\rho$ between the denominator of full and sparse attention according to Eq. 8. Because we compute full attention in the caching step, the ratio $\rho$ is not an approximation but an *accurate* value. Hence, we have:

$$
\text{ReshapeAttention}(Q, K, V, \rho) = \rho A \cdot V = A_{\in S} \cdot V
\tag{12}
$$

Therefore, the residual $R$ in Eq. 10 is:

$$
R = A \cdot V - A_{\in S} \cdot V = A_{\notin S} \cdot V,
\tag{13}
$$

which is mathematically equivalent to the attention output of the masked tokens at the caching timestep.

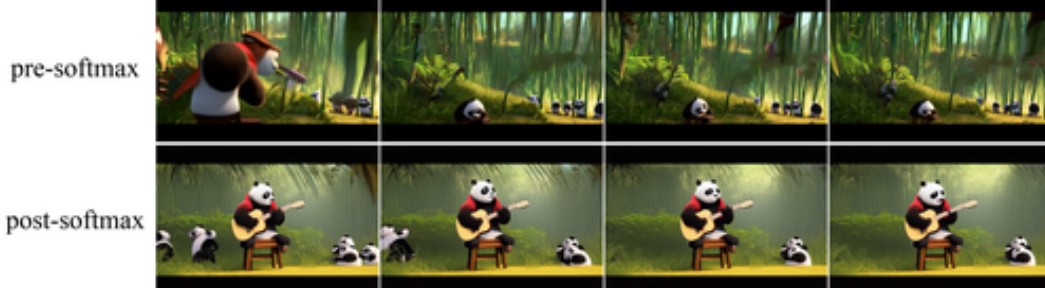

Figure 7: Visual comparison of pre-softmax and post-softmax masking on CogVideoX-2B with 66% sparsity. Prompt: "A panda, dressed in a small, red jacket and a tiny hat, sits on a wooden stool in a serene bamboo forest. The panda's fluffy paws strum a miniature acoustic guitar, producing soft, melodic tunes. Nearby, a few other pandas gather, watching curiously and some clapping in rhythm. Sunlight filters through the tall bamboo, casting a gentle glow on the scene. The panda's face is expressive, showing concentration and joy as it plays. The background includes a small, flowing stream and vibrant green foliage, enhancing the peaceful and magical atmosphere of this unique musical performance".

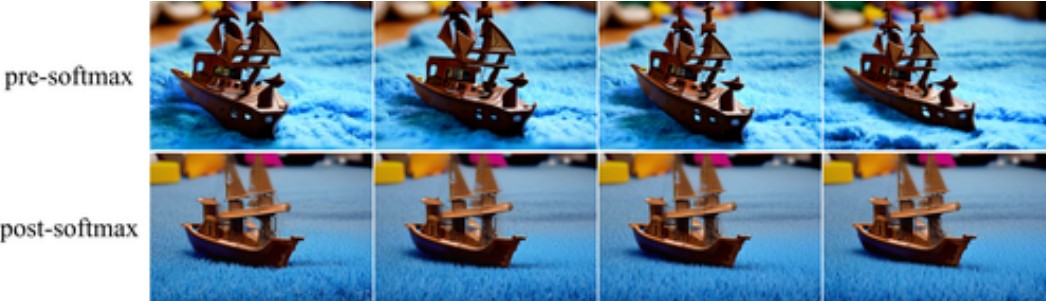

Figure 8: Visual comparison of pre-softmax and post-softmax masking on CogVideoX-2B with 66% sparsity. Prompt: "A detailed wooden toy ship with intricately carved masts and sails is seen gliding smoothly over a plush, blue carpet that mimics the waves of the sea. The ship's hull is painted a rich brown, with tiny windows. The carpet, soft and textured, provides a perfect backdrop, resembling an oceanic expanse. Surrounding the ship are various other toys and children's items, hinting at a playful environment. The scene captures the innocence and imagination of childhood, with the toy ship's journey symbolizing endless adventures in a whimsical, indoor setting".

## A.4 Details of Baseline Implementation

We compare Re-ttention to three different baseline methods: Sparse VideoGen (SVG) [42], MInference [18], and DiTFastAttn [48]. We enumerate the experiment implementation details for Text-to-Video (T2V) and Text-to-Image (T2I) generation tasks, respectively.

**Text-to-Video**    For DiTs, MInference classifies all attention heads into a block sparse format [18] to generate $M$. We use the SVG official implementation for CogVideoX series to generate videos. As for Re-ttention and DiTFastAttn, we use sliding window attention, which restricts each token's attention to a local neighborhood and will repeat the same mask at each frame of the video.

**Text-to-Image**    Since T2I generation lacks a temporal dimension, we apply only the spatial attention heads in SVG and adjust the window size to match the target sparsity. Re-ttention and DiTFastAttn use the same sliding window attention as SVG.

## A.5 Additional Examples for Post-Softmax Masking Operation

Figure 7 expands on Fig. 4 by providing additional video frame comparisons and the lengthy textual prompt. Post-Softmax masking not only better preserves the objects (panda, stool, guitar, etc.), but also consistently maintains the main part of the video over time, while pre-softmax causes the

large panda to vanish. This example further validates our assumption regarding the importance of maintaining the softmax denominator.

Further, Figure 8 provides an additional comparison with a different prompt. In the pre-softmax video, the ship becomes increasingly distorted over time, whereas in the post-softmax video, it remains consistent throughout. Notably, the reduced texture detail in the post-Softmax output reveals an issue caused by denormalized attention probabilities—specifically, information loss due to the sum of softmax probabilities being less than one, leading to a shrinkage effect in the features.

## A.6 Visual Comparison for Video Generation

We show additional visual (qualitative) comparisons on video generation using the CogVideoX-2B [47] model in Figures 9, 10, 11 and 12. For example, in Figure 10, Re-ttention has the best looking otter as well as the food with the most similar shape as the reference video. Besides, while other baseline methods have artifacts like blurry textures and distortions in the background, Re-ttention preserves background fidelity, closely matching the reference video. Those additional comparisons match the experiment result in the main paper: The videos generated by Re-ttention are the most similar to the reference video generated by full-attention; also, it has the best clarity and consistency and no artifacts in the background.

## A.7 Visual Comparison for Image Generation

We show additional visual (qualitative) comparisons on image generation using the PixArt-$\alpha$ [3], PixArt-$\Sigma$ [2], and Hunyuan [23] models in Figures 13, 14 and 15, respectively. The main object generated by the dynamic sparse attention baseline MInference deviates significantly from the full-attention reference, often resulting in unnatural or distorted appearances. For static baseline methods like SVG and DiTFastAttn, although the main objects in their images are more similar to the full-attention reference images, there are artifacts in the background which degrade the image quality. In comparison, Re-ttention not only preserves the fidelity of the main object but also mitigates background artifacts, demonstrating superior performance in T2I generation and strong generalization across different DiT architectures.

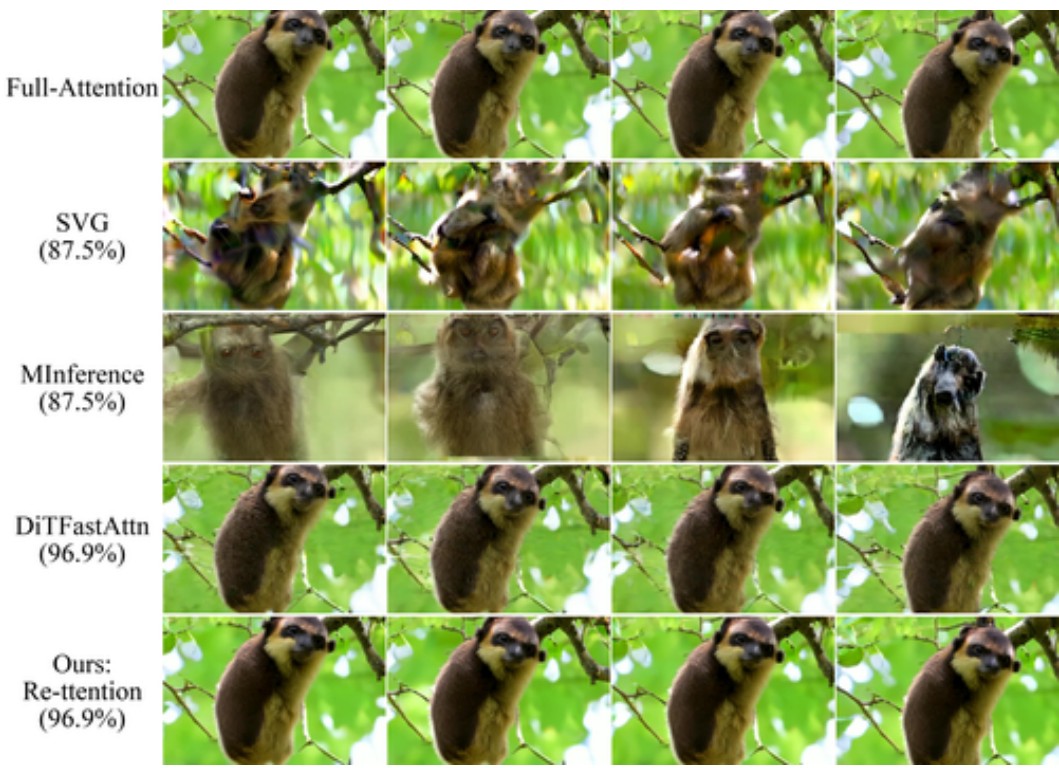

Figure 9: **T2V** visual comparison using CogVideoX-2B [47] T2V model. Each row corresponds to video frames generated by different methods. Prompt: "a curious sloth hanging from a tree branch".

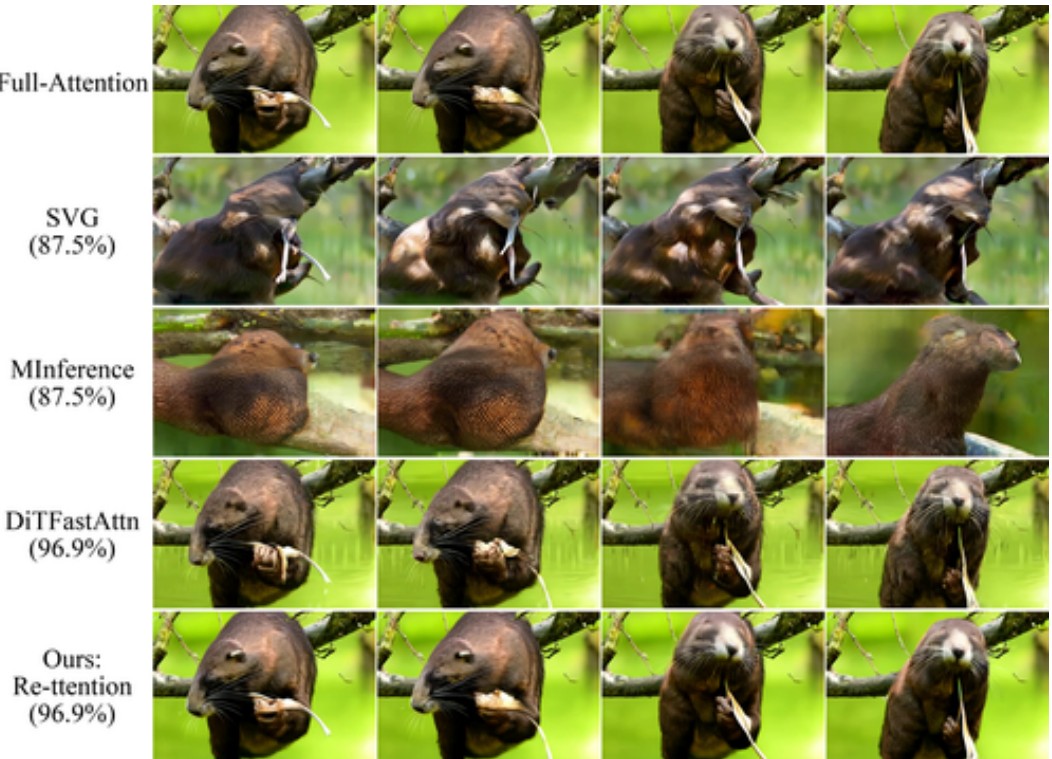

Figure 10: **T2V** visual comparison using CogVideoX-2B [47] T2V model. Each row corresponds to video frames generated by different methods. Prompt: "otter on branch while eating".

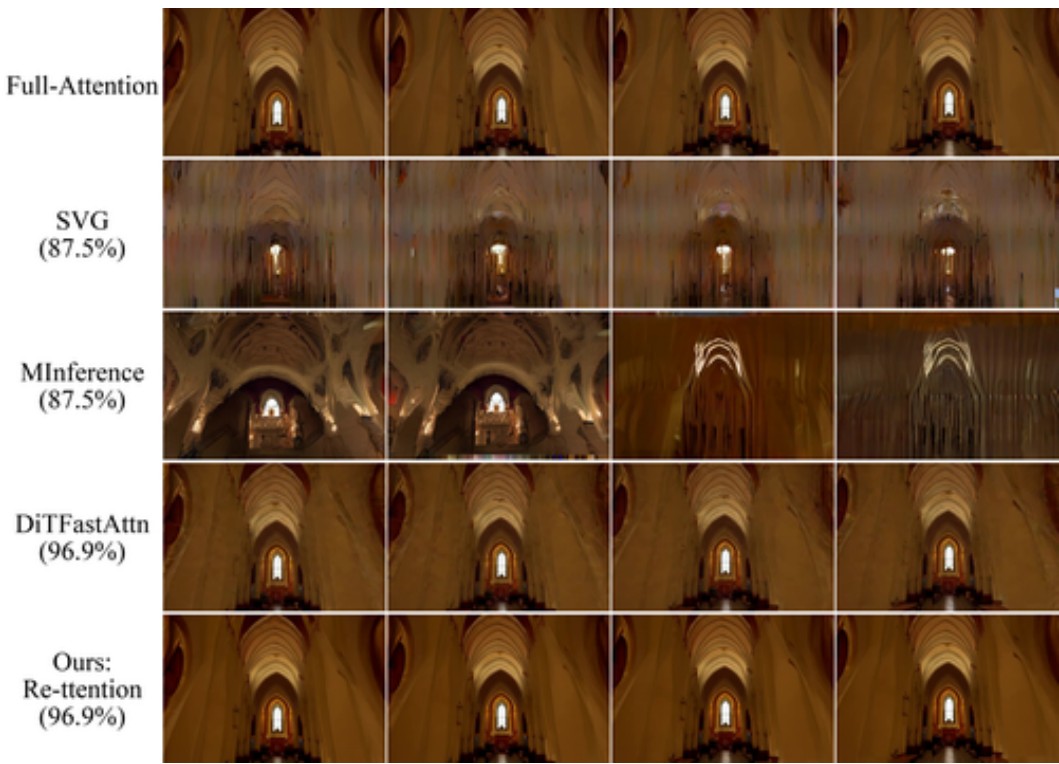

Figure 11: **T2V** visual comparison using CogVideoX-2B [47] T2V model. Each row corresponds to video frames generated by different methods. Prompt: "a church interior".

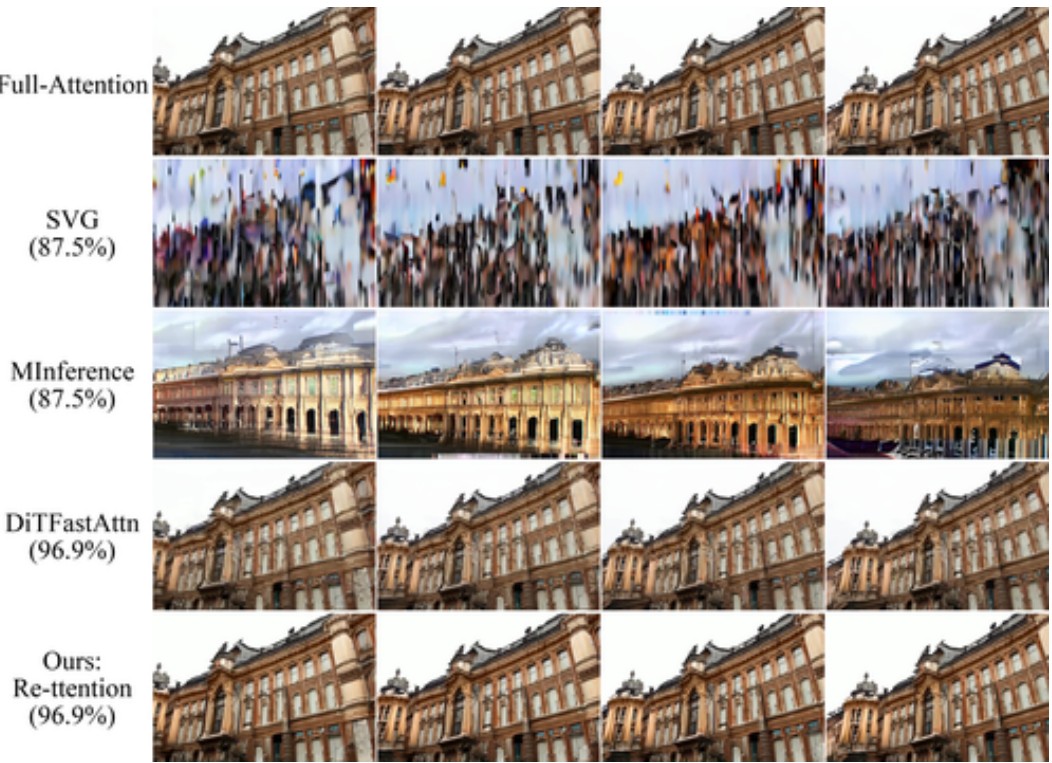

Figure 12: **T2V** visual comparison using CogVideoX-2B [47] T2V model. Each row corresponds to video frames generated by different methods. Prompt: "the georgian building".

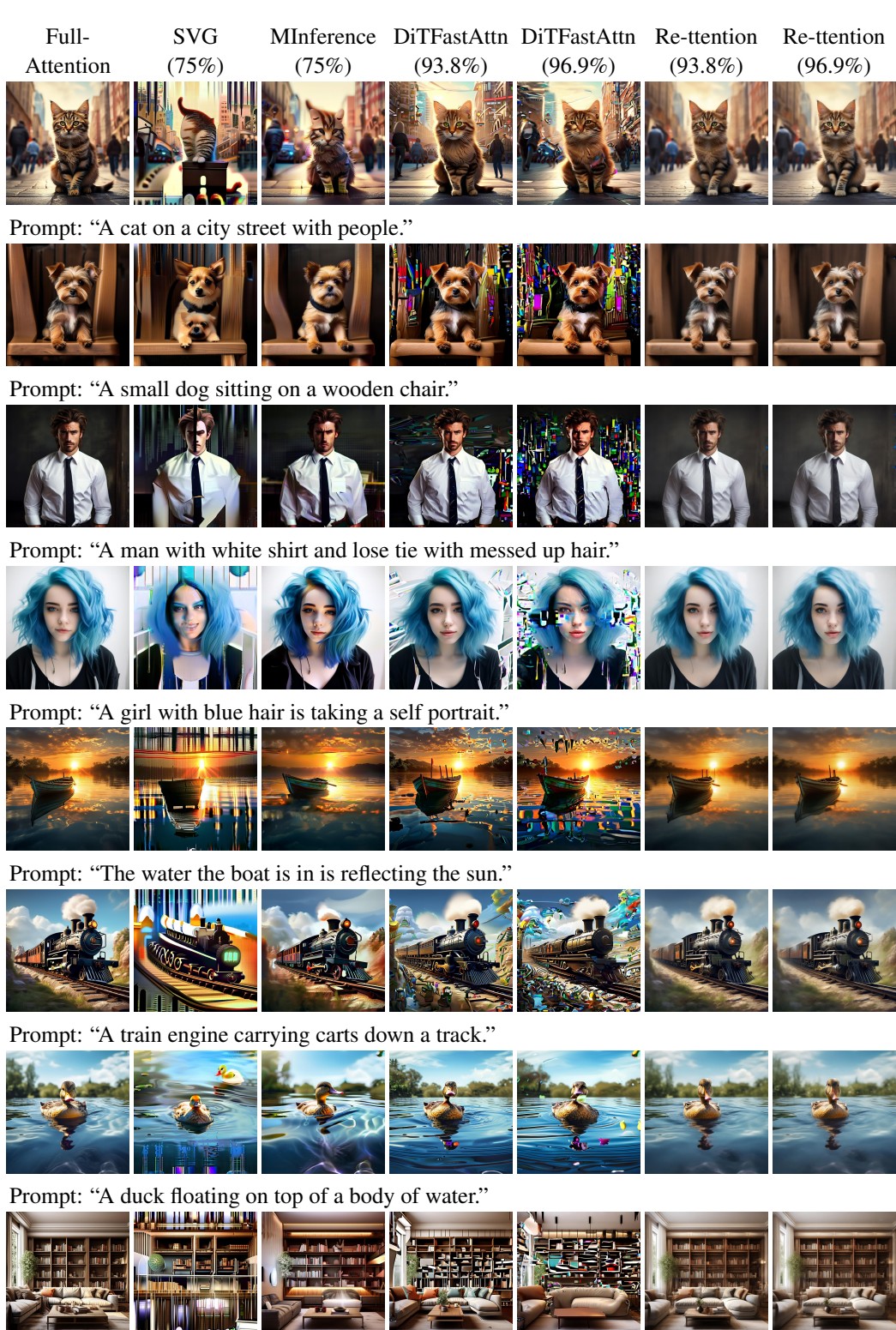

Figure 13: **T2I** visual comparison on MS-COCO 2014 [26] dataset using PixArt-$\alpha$ [3] model. Each row corresponds to one prompt, we show images generated by Re-ttention (our method) and by other attention methods in different columns.

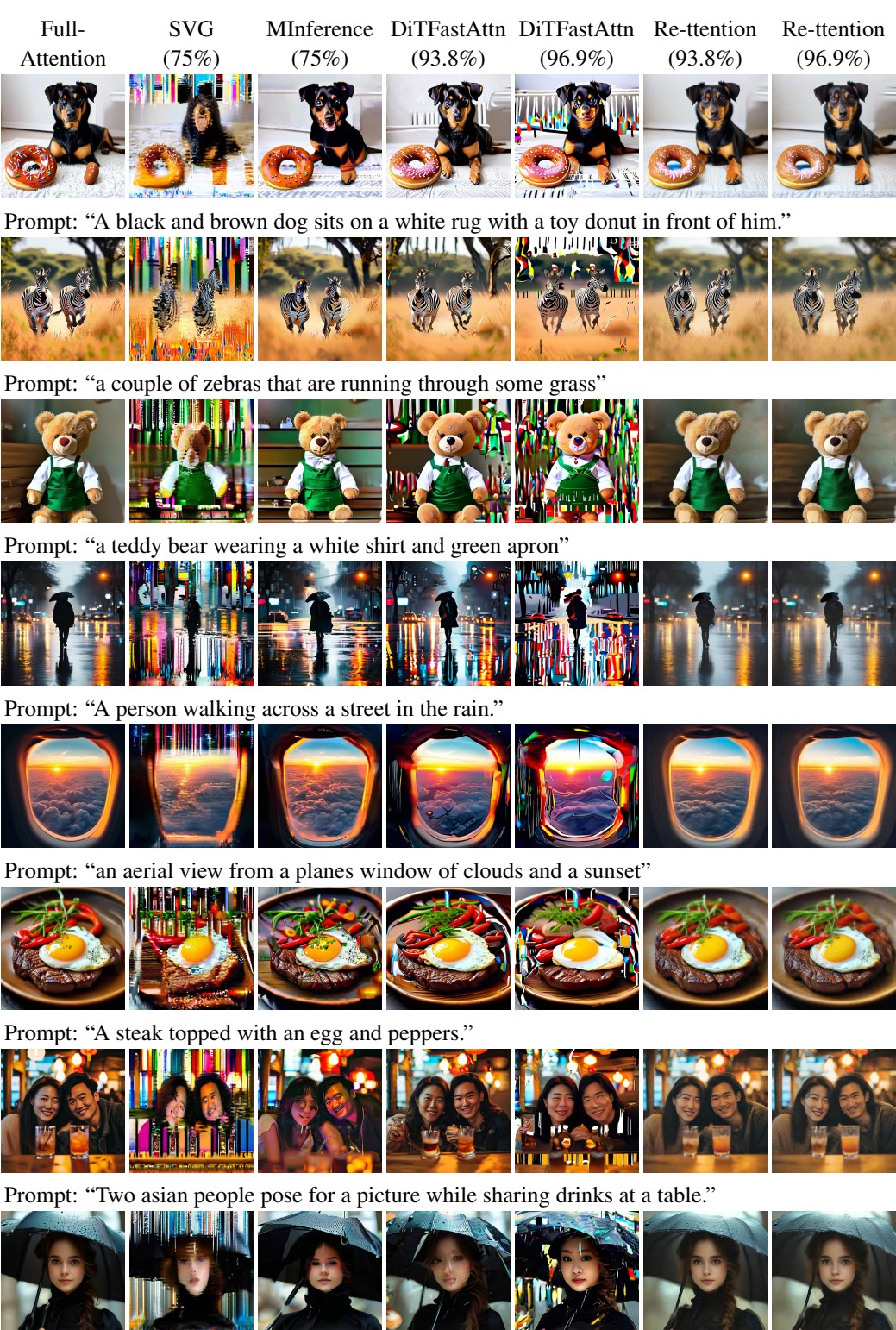

|  Full-Attention | SVG (75%) | MInference (75%) | DiTFastAttn (93.8%) | DiTFastAttn (96.9%) | Re-ttention (93.8%) | Re-ttention (96.9%) |

Prompt: "A black and brown dog sits on a white rug with a toy donut in front of him."

Prompt: "a couple of zebras that are running through some grass"

Prompt: "a teddy bear wearing a white shirt and green apron"

Prompt: "A person walking across a street in the rain."

Prompt: "an aerial view from a planes window of clouds and a sunset"

Prompt: "A steak topped with an egg and peppers."

Prompt: "Two asian people pose for a picture while sharing drinks at a table."

Prompt: "A pretty young lady holding a black umbrella."

Figure 14: **T2I** visual comparison on MS-COCO 2014 [26] dataset using PixArt-$\Sigma$ [2] model. Each row corresponds to one prompt, we show images generated by Re-ttention (our method) and by other attention methods in different columns.

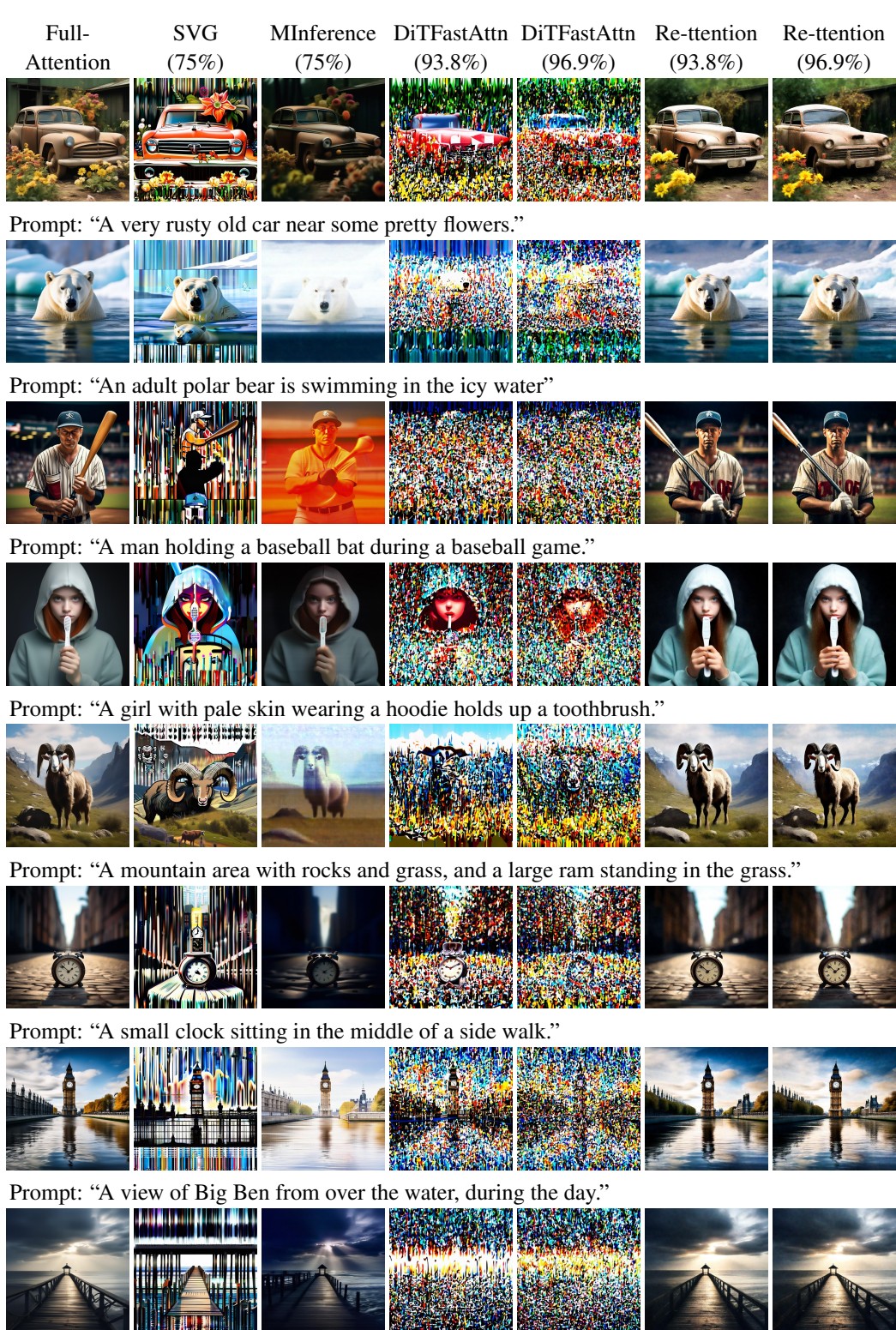

| Full-Attention | SVG (75%) | MInference (75%) | DiTFastAttn (93.8%) | DiTFastAttn (96.9%) | Re-ttention (93.8%) | Re-ttention (96.9%) |

Prompt: "A very rusty old car near some pretty flowers."

Prompt: "An adult polar bear is swimming in the icy water"

Prompt: "A man holding a baseball bat during a baseball game."

Prompt: "A girl with pale skin wearing a hoodie holds up a toothbrush."

Prompt: "A mountain area with rocks and grass, and a large ram standing in the grass."

Prompt: "A small clock sitting in the middle of a side walk."

Prompt: "A view of Big Ben from over the water, during the day."

Prompt: "Light breaks through a cloudy day at the pier."

Figure 15: **T2I** visual comparison on MS-COCO 2014 [26] dataset using Hunyuan-DiT [23] model. Each row corresponds to one prompt, we show images generated by Re-ttention (our method) and by other attention methods in different columns.

