# OpenReview forum: "Re-ttention: Ultra Sparse Visual Generation via Attention Statistical Reshape"
_NeurIPS.cc/2025/Conference — NeurIPS 2025 poster_

### Official Review · Reviewer_phiV · 2025-07-01

**Clarity:** 3
**Significance:** 2
**Originality:** 2
**Rating:** 4
**Confidence:** 4

**Summary:**

This paper proposes Re-ttention, a training-free sparse attention mechanism for Diffusion Transformers that enables ultra-sparse inference (up to 96.9% sparsity) without quality degradation. It addresses the softmax distribution shift caused by sparsity by reusing attention score statistics across denoising steps via a cached denominator ratio and residual correction. The method is simple to implement, incurs negligible overhead, and achieves over 92% self-attention latency reduction. Extensive experiments on T2I and T2V tasks demonstrate that Re-ttention outperforms prior sparse attention baselines like DiTFastAttn, MInference, and SVG in both quality and efficiency.

**Questions:**

1. How sensitive is the method to the static attention mask design? Since Re-ttention relies on fixed sparse masks, how would it perform under dynamically computed masks (e.g., block sorting)? Could the authors clarify whether the denominator approximation remains valid when token positions change?

2. Is the ρ-ratio globally or locally applied? In Section 4.2, the paper defines ρ as a per-token ratio, but in practice it is unclear whether ρ is shared across heads, tokens, or averaged. A clarification and analysis of how local vs. global scaling affects quality would be helpful.

3. Error accumulation in long denoising schedules: Re-ttention reuses ρ across denoising steps, adding a ramp-up λ. Could this accumulate estimation errors over time? Have the authors tested performance for longer denoising schedules (e.g., 100+ steps) or varied sampling schedules?

4. Is residual caching practical under memory constraints? While self-attention latency is reduced, the method introduces a memory-time tradeoff by caching full attention outputs periodically. Could the authors provide quantitative memory overhead analysis?

5. Does Re-ttention generalize to conditional control tasks? For example, text-to-image with spatial constraints or layout conditions. Can the statistical correction mechanism interfere with structured control, or can it be combined with existing conditioning modules?

**Ethical Concerns:**

["NO or VERY MINOR ethics concerns only"]

**Final Justification:**

The authors’ detailed response has satisfactorily addressed most of my concerns, which encouraged me to raise my score.

**Limitations:**

The paper briefly discusses limitations in future work (e.g., extending to LLMs and dynamic masks), but does not thoroughly analyze potential risks such as mismatch in attention semantics, degradation under noisy prompts, or applicability to constrained generation tasks.

**Paper Formatting Concerns:**

NaN

**Quality:**

2

**Strengths And Weaknesses:**

**Strengths**

1. The paper proposes a simple yet effective method to mitigate softmax distribution shift in sparse attention via cached normalization and residual correction.

2. Re-ttention achieves ultra-high sparsity (96.9%) with minimal quality loss and over 92% self-attention latency reduction across T2I and T2V tasks.

3. The method is training-free, generalizable across multiple DiT models, and thoroughly validated with strong empirical results.

**Weakness**

1. The statistical correction introduced by the ρ-scaling and residual caching lacks a theoretical grounding—there is no guarantee that the modified attention remains consistent with probabilistic interpretations or converges to full attention.

2. While the method is empirically validated across several models, the sparse mask remains static. This limits generalization to dynamic attention scenarios, and the paper does not examine how Re-ttention interacts with token-level adaptivity or downstream control tasks.

3. The ablation is relatively narrow: only one hyperparameter (λ) is tested on a subset of benchmarks, without broader sensitivity analysis or exploration of failure modes.

---

> ### Author Rebuttal · Authors · 2025-07-31
>
> >**W1: The statistical correction introduced by the ρ-scaling and residual caching lacks a theoretical grounding—there is no guarantee that the modified attention remains consistent with probabilistic interpretations or converges to full attention.**
>
> We offer a better approximation to full attention under ultra-sparse settings than prior works under a new assumption, emphasizing the importance of denominator rescaling. The residual caching in existing sparse attention methods can not guarantee convergence to full attention and in fact is relying on some assumption that may not hold. For example, given a row of true full attention scores: $\text{softmax}(QK) = [0.1, 0.4, 0.4, 0.1]$, DiTFastAttn computes the sparse attention scores as $\text{softmax}(QK) = [0, 0.5, 0.5, 0]$ and the residual as $R = \text{full attention - sparse attention} =  [0.1, -0.1, -0.1, 0.1]$ via subtraction. DiTFastAttn’s residual contaminates the attention scores if the attention distribution inside the window (the middle two points) changes, i.e., given full attention $\text{softmax}(QK) = [0.1, 0.3, 0.5, 0.1]$, the sparse attention is $\text{softmax}(QK) = [0, 0.375, 0.625, 0]$. Reconstructed with residual: [0.1, 0.275, 0.525, 0.1], which is inaccurate in the sparse window. Our method rescales the sparse attention scores as $\text{softmax}(QK) = [0, 0.4, 0.4, 0]$ and computes the residual as $R’ = [0.1, 0, 0, 0.1]$. Even if the full attention changes to $[0.1, 0.3, 0.5, 0.1]$ and the sparse attention is $[0, 0.375, 0.625, 0]$, after applying our proposed statistical reshaping, it becomes $[0, 0.3, 0.5, 0]$. Adding the residual $R'$ makes it match the full attention. That is to say, as long as the $\rho$ defined by Eq. 8 is estimated accurately and residual attention outside the window does not change much, we approximate the full attention correctly.
>
> >**W2: While the method's sparse mask remains static. This limits generalization to dynamic attention scenarios, and the paper does not examine how Re-ttention interacts with token-level adaptivity or downstream control tasks.**
>
> The focus of this paper is on static sparse attention, as it has been shown that static masks are a solid choice for implementing sparse attention, especially in the context of T2I/T2V Diffusion Models, where the generation process involves spatially or temporally structured visual tokens [1]. Compared to the dynamic mask, the static mask is easier to implement and introduces no overhead during inference. However, the statistical reshape for the dynamic mask is worth studying in future work, which is beyond our focus in this paper. That being said, we perform some tentative exploration on adapting Re-ttention on the dynamic sparse mask, please refer to the result shown in response to Q1. As for downstream control tasks, please refer to Q5's response.
>
> >**W3: The ablation is relatively narrow.**
>
> We conduct more ablation in terms of sampling schedule (see Q.3) and applying dynamic mask (see Q.1).
>
> >**Q1: How sensitive is the method to the static attention mask design? Since Re-ttention relies on fixed sparse masks, how would it perform under dynamically computed masks? Could the authors clarify whether the denominator approximation remains valid when token positions change?**
>
> 1. Based on our observations through tentative exploration (which is beyond the focus of this work), the conclusion we make about the denominator approximation still holds for dynamically computed masks, as the ratio ρ exhibits a predictable trend even with dynamically calculated masks. However, because the masking pattern changes at each step, the cached residual can not correctly represent the attention output of masked tokens, thereby resulting in suboptimal performance.
> 2. We provide experiment results on directly applying Re-ttention to PixArt-$\Sigma$ with dynamically calculated masks. The results show that the performance drops after replacing the static mask with the dynamic mask. Note that dynamically calculated masks impose additional computation compared to static attention mask. That being said, exploration of dynamic masks is subject to future investigations, as this paper focuses on maintaining the quality of sparse window attention (with static masks) for DiT.
>
> | Attention              | HPSv2 ↑ | GenEval ↑ |
> |--|--|--|
> | Static mask attention  | 27.72   | 0.512      |
> | Dynamic mask attention | 25.35   | 0.473      |
>
> >**Q2: Is the ρ-ratio globally or locally applied? In Section 4.2, the paper defines ρ as a per-token ratio, but in practice it is unclear whether ρ is shared across heads, tokens, or averaged. A clarification and analysis of how local vs. global scaling affects quality would be helpful.**
>
> The ρ-ratio is locally applied per attention head, as we described in Section 4.2. More specifically, in Fig. 5, we show the ρ-ratio for a single token at a given attention head. In our observations, the ρ-ratio for a specific token varies greatly across different attention heads, and inaccurately predicting the scaling ρ with an averaged value will be harmful to the performance. Thus, we use a local scaling mechanism.
>
> >**Q3: Error accumulation in long denoising schedules: Re-ttention reuses ρ across denoising steps, adding a ramp-up λ. Could this accumulate estimation errors over time? Have the authors tested performance for longer denoising schedules (e.g., 100+ steps) or varied sampling schedules?**
>
> Yes, the estimation error can accumulate over time when reusing ρ across denoising steps. To address this, we periodically insert full attention steps, which effectively act as a reset mechanism to correct any accumulated estimation error in the residuals. In the following, we further add the result on PixArt-$\Sigma$ with 96.9% sparsity after changing the sampling schedule from $T=5$ to 3 and 7. Smaller $T$ means more full attention inferences and more accurate cached residual, thereby resulting in better performance. However, our Re-ttention outperforms DiTFastAttn regardless of $T$. We believe that the result demonstrates the effectiveness of our method across different sampling schedules.
>
> | Attention     | T = 3 GenEval ↑ | T = 3 HPSv2 ↑ | T = 5 GenEval ↑ | T = 5 HPSv2 ↑ | T = 7 GenEval ↑ | T = 7 HPSv2 ↑ |
> |---|---|---|---|---|---|---|
> | DiTFastAttn   | 0.376            | 26.81          | 0.233            | 22.79          | 0.147            | 20.87          |
> | Re-ttention   | 0.544            | 29.26          | 0.512            | 27.72          | 0.479            | 26.16          |
>
> >**Q4: Is residual caching practical under memory constraints? While self-attention latency is reduced, the method introduces a memory-time tradeoff by caching full attention outputs periodically. Could the authors provide quantitative memory overhead analysis?**
>
> 1. Yes, it is practical and under memory constraints. In most DiT architectures [2], since the cached residual has the same shape as the attention output, its memory overhead is typically considered negligible relative to the model weights. We use PixArt-$\Sigma$ as an example to perform a quantitative analysis on the memory consumption of the cached residual. We use the HuggingFace implementation to measure the memory consumption. Loading PixArt-$\Sigma$ model takes 13080MiB, while the cached residual (for one 1024x1024 image) is 36 MiB, which is ignorable.
> 2. Furthermore, it involves only simple addition and subtraction operations, which are memory-efficient and impose minimal computational burden [3].
>
> >**Q5: Does Re-ttention generalize to conditional control tasks? Can the statistical correction mechanism interfere with structured control, or can it be combined with existing conditioning modules?**
>
> 1. Yes, Re-ttention is applicable to the conditional control tasks as well. This potential has already been shown in our T2V experiment, which uses CogVideoX model and is a scenario that needs to handle conditional control tokens (i.e., text tokens as control tokens). This is because CogVideoX model follows the MMDiT architecture [4]. This architecture omits explicit cross-attention and instead employs a unified self-attention mechanism over both visual tokens and conditional tokens (including text, image, layout tokens). In Table 1, we can see that Re-ttention can adapt to this setting and achieves outstanding performance, outperforming other baseline methods across 5 different metrics related to quality, similarity, and consistency.
> 2. Another setting of conditional control tasks involves cross-attention between control tokens and generated visual tokens, which is beyond the scope of this work. The focus of this paper is on ultra-sparse self-attention, since sparse attention is usually applied to self-attention, i.e., the bulk of attention computation cost is for the self-attention mechanism rather than cross-attention. In fact, it is common practice in the literature [5] to apply full attention to cross-attention layers, as the computational cost of cross-attention is relatively low, making sparse attention unnecessary.
>
> [1] H. Xi, S. Yang, Y. Zhao, et al., “Sparse VideoGen: Accelerating video diffusion transformers with spatial-temporal sparsity,” arXiv preprint arXiv:2502.01776, 2025.
>
> [2] W. Peebles and S. Xie, “Scalable diffusion models with transformers,” in Proc. IEEE/CVF Int. Conf. Comput. Vis. (ICCV), 2023, pp. 4195–4205.
>
> [3] T. Dao, D. Fu, S. Ermon, et al., “FlashAttention: Fast and memory-efficient exact attention with IO-awareness,” Adv. Neural Inf. Process. Syst. (NeurIPS), vol. 35, pp. 16344–16359, 2022.
>
> [4] P. Esser, S. Kulal, A. Blattmann, et al., “Scaling rectified flow transformers for high-resolution image synthesis,” in Proc. 41st Int. Conf. Mach. Learn. (ICML), 2024.
>
> [5] Z. Yuan, H. Zhang, L. Pu, X. Ning, et al., “DiTFastAttn: Attention compression for diffusion transformer models,” Adv. Neural Inf. Process. Syst. (NeurIPS), vol. 37, pp. 1196–1219, 2024.

---

> ### Comment · Reviewer_phiV · 2025-08-04
> **Respones to Authors**
>
> Thank you for the authors’ response, which has addressed most of my concerns and motivated me to **increase my score**. However, I remain very curious about the performance of Re-ttention in conditional generation tasks based on MM-DiT modules, such as OminiControl. Since these architectures concatenate condition tokens with visual tokens for self-attention, they tend to significantly increase computational complexity. If Re-ttention can demonstrate strong performance in such conditional generation settings, it would greatly enhance the contribution and impact of this work to the community.

---

> > ### Author Response · Authors · 2025-08-06
> >
> > We thank the reviewer for their thoughtful feedback. As noted by the reviewer, a current trend in the development of Diffusion Transformers (DiTs) favors a unified self-attention mechanism over catenated visual tokens and other tokens (including conditional control tokens) instead of cross-attention, which exacerbates the computation cost of attention. To tackle the ever-increasing bottleneck of self-attention of longer and concatenated tokens, this work aims to establish the foundation and techniques of achieving ultra sparse attention on general DiTs while maintaining the quality (i.e., achieving 96.9% sparsity for the first time in the literature). We perform extensive evaluation on DiT tasks including T2I and T2V (where CogVideoX is based on the MM-DiT architecture).
> >
> > Although specific downstream tasks are not the theme of this paper, we provide some explanation on the possibility of applying Re-ttention to conditional generation tasks like OminiControl [1]. Ominicontrol considers two types of control tasks, which are “spatially-aligned” and “non-aligned” control tasks:
> >
> > 1. For “spatially-aligned” control tasks (like canny-to-image, depth-to-image), the tokens in conditional image have a spatial correspondence to those in the image to be generated. Thus, it is feasible to extend Re-ttention to handle these cases. For example, Figure 4a in [1] shows the attention map for canny-to-image tasks, featuring three diagonal lines on the attention map. The attention mask used by Re-ttention already covers the central diagonal, and it is feasible to extend it to include other spatially defined masks in self-attention.
> > 2. For “non-aligned” control tasks (like subject driven generation), Re-ttention may not be applied with ease. The attention behaviour between noisy image tokens and image condition tokens (the upper-right corner and lower-left corner of Figure 4a in [1]) may not follow a static spatial pattern but rather depend on the specific input condition. For example, Figure 4b in [1] demonstrates the attention map for subject driven generation, where the attention pattern resembles the outline of a chicken in the given input condition. Designing sparse attention that can effectively capture highly dynamic patterns in the conditional input still deserves much investigation.
> >
> > That being said, this work mainly focuses on enabling ultra-sparse attention for DiT, evaluated on a range of representative T2I and T2V models and benchmarks. We leave the thorough investigation of applicability to conditional generation tasks to future work.
> >
> > [1] Tan, Zhenxiong, et al. "Ominicontrol: Minimal and universal control for diffusion transformer." arXiv preprint arXiv:2411.15098 (2024).

---

### Official Review · Reviewer_SARk · 2025-07-02

**Clarity:** 3
**Significance:** 3
**Originality:** 3
**Rating:** 4
**Confidence:** 4

**Summary:**

This paper presents a method for sparse attention in image and video generation. It re-scales the attention matrix of sparse attention to be close to full attention. It also adds residual from a previous layer to the following attent results. The method has been applied to text-to-image and text-to-video generation models. The proposed method shows better evaluation metrics when compared with previous methods.

**Questions:**

-  Is the degradation of DiTFastAttn on PixArt-Sigma related to token compression? How does Re-ttention handle this change?

**Ethical Concerns:**

["NO or VERY MINOR ethics concerns only"]

**Final Justification:**

The authors addressed my concerns on experimental results and the significance. Although the proposed attention scaling is a small change to previous work, it's effective to improve attention sparsity from 70% to 95%. Additional experiments show that 95% sparsity has a significant impact to computation speed.

**Limitations:**

Yes

**Quality:**

3

**Strengths And Weaknesses:**

Strength:
- The method is training-free, and can be applied to any diffusion models with attentions.
- The method shows better metrics in evaluations.

Weakness:
- The paper brings limited novelty. Residual caching was introduced by DiTFastAttn. Apart from that, the contribution of this paper is the attention scaling, which is minor.
- Unclearness about experimental results. For T2V, the proposed method is slightly better than previous method DiTFastAttn. However for T2I (Figure 6), DiTFastAttn performs well on PixArt-alpha, but severely degrades on other methods. The paper didn’t analyze the degradation.
- In ablation studies, the paper only studies hyperparameter lambda.  In addition to that, the paper should ablate the components of the proposed method: attention rescaling and residual caching.
- Writing can be improved.

---

> ### Author Rebuttal · Authors · 2025-07-31
>
> >**W1. The paper brings limited novelty. Residual caching was introduced by DiTFastAttn. Apart from that, the contribution of this paper is the attention scaling, which is minor.**
>
> We would like to invite the reviewer to re-evaluate the value and significance of Re-ttention through the aspects of 1) the limitation of prior sparse attention and caching method for DiT and why these schemes cannot uphold their statistical assumption; 2) how our revised statistical assumption fills the important gap in DiT attention approximation, and the mathematically grounded statistical reshaping method in Re-ttention to cope with attention distribution drift that supports our assumption; 3) the significant performance improvement over existing schemes on both T2I and T2V tasks to achieve much higher sparsity levels that the literature has never reported.
> We respectfully disagree with this comment, which might be a misunderstanding or at least an over-simplified/downgraded view of our contribution. In the following, we explain the key differences in detail:
>
> 1. DiTFastAttn’s failed caching assumption: Sparse attention with residual caching in DiTFastAttn caches the difference (subtraction) between full and sparse attention in one step and uses this difference in subsequent steps, based on the assumption that the “residual” changes minimally from step to step. However, this assumption is not true, since it naturally assumes the attention scores of tokens within the window remain unchanged across timesteps. For example, given a row of full attention scores: $\text{softmax}(QK) = [0.1, 0.4, 0.4, 0.1]$, DiTFastAttn computes the sparse attention scores as $\text{softmax}(QK) = [0, 0.5, 0.5, 0]$ and the residual as $R = \text{full attention - sparse attention} =  [0.1, -0.1, -0.1, 0.1]$. If in the next step the true full attention scores have a different distribution inside the window (the 2 middle positions), e.g. $\text{softmax}(QK) = [0.1, 0.3, 0.5, 0.1]$, the sparse attention output becomes $\text{softmax}(QK) = [0, 0.375, 0.625, 0]$. The attention reconstructed with the cached residual $R = [0.1, -0.1, -0.1, 0.1]$ is $[0.1, 0.275, 0.525, 0.1]$, which is incorrect in the sparse window (the most important part for attention). In addition to the abuse of this assumption, the common limitation of existing sparse attention methods, including DiTFastAttn, is that they ignore the importance of the denominator in softmax operation.
>
> 2. Re-ttention’s new assumption, statistical reshaping and caching schemes: Re-ttention fixes this issue by reconstructing softmax denominator by a cached $\rho$ (as detailed in Sec. 4.2) together with our specific residual caching for attention only outside the window. In other words, our corrected mathematical assumption is that attention outside the window changes minimally across steps, whereas sparse attention within the window is critical and could change from step to step and thus demands a whole new reshaping technique to reconstruct the full attention. Specifically, given the same row of attention scores $\text{softmax}(QK) = [0.1, 0.4, 0.4, 0.1]$, Re-ttention caches [0.1, 0, 0, 0.1] as the residual, i.e., only the attention of masked tokens outside the sparse window (see Appendix A.3). In the next timestep, if $\text{softmax}(QK) = [0.1, 0.3, 0.5, 0.1]$, the sparse attention output is $\text{softmax}(QK) = [0, 0.375, 0.625, 0]$. After applying our proposed statistical reshaping, it becomes $[0, 0.3, 0.5, 0]$ (Please refer to Sec. 4.2 for how to reshape the sparse attention). Adding the residual attention outside the window $[0.1, 0, 0, 0.1]$ makes it match the full attention. Thus, we provide a mathematical grounded solution: as long as the ratio $\rho$ as defined by Eq (8) is estimated accurately and residual attention outside the window does not change much, there is a good chance of approximating the full attention, even if attention distribution changes within the window. However, with the prior caching method in DiTFastAttn, we can see that the reconstruction is almost always wrong if the detailed distribution of sparse attention within the sparse window alters from time step to step, which happens frequently if a high sparsity level is applied (leading to even smaller windows).
>
> 3. Experimental evidence: We largely surpass all the baseline methods in both quantitative and qualitative performance evaluation on multiple benchmarks and tasks. In quantitative metrics, Re-ttention outperforms DiTFastAttn across all T2V and T2I tasks. Especially in the T2I tasks, Re-ttention with 96.9% sparsity outperforms DiTFastAttn with 93.8% sparsity in Table 2, across 5 different benchmarks (HPSv2, GenEval, LPIPS, IR, CLIP). For qualitative visual comparison, Re-ttention preserves the visual quality and maintains high consistency even under 96.9% sparsity, while DiTFastAttn suffers from visual artifacts, degraded textures, and even semantic corruption in Figures 1, 6, 9, 10, 11, 12, 13, 14, 15.  Such a significant gap across a wide range of benchmarks and metrics would not have been possible if Re-ttention is a minor increment of prior sparse attention work.
>
> In summary, Re-ttention is a holistic mathematically grounded method to revisit how full attention can be recovered from sparse attention through caching in DiT, and requires two necessary parts to work together: statistical reshaping of attention within the sparse window via the estimated denominator, and residual compensation by caching masked tokens’ attention.
>
>
> >**W2. Unclearness about experimental results. For T2V, the proposed method is slightly better than previous method DiTFastAttn. However for T2I (Figure 6), DiTFastAttn performs well on PixArt-α, but severely degrades on other methods. The paper didn’t analyze the degradation.**
>
> We would like to note that all the results are authentically reported by following the implementation details provided in the original papers or by executing their publicly available codes, and analyzing the incompetence of other baseline methods is out of the main focus of this paper. However, we provide some plausible explanations for these results regarding DiTFastAttn.
> 1. The more severe degradation observed in DiTFastAttn is primarily due to different latent spaces. PixArt-α uses the LDM’s visual autoencoder (VAE), while PixArt-$\Sigma$ and Hunyuan use SDXL’s VAE. SDXL’s VAE uses the same architecture as LDM’s, but has superior performance due to the improved training procedure. This makes it more expressive, but likely more sensitive to minor changes in latent distribution. Due to the distribution shift of the attention score, DiTFastAttn deviates from the full attention at ultra-sparse settings, which causes varying degrees of degradation—in Sigma and Hunyuan, the degradation is more severe as the latent space is more sensitive to the deviation.
> 2. We use the visual result in Figure 6 to explain our claim above. In PixArt-α, the latent space is more robust, and distribution shifts can lead to minor changes, although still noticeable, in semantic attributes such as the breed of the cat and the style or quality of the background (although there are still some visible artifacts in certain areas). In the other two models, which use more sensitive latent spaces, the impact of distribution shifts due to sparse attention is more severe—resulting in prominent visual artifacts, degraded textures, and even semantic corruption such as distorted object shapes or inconsistent attributes. In contrast, Re-ttention does not suffer from the same degree degradation—it preserves the visual quality and maintains high consistency under ultra sparse settings. Additional visual comparison can be found in Figure 13, 14, 15.
> 3. Literature shows that compressing PixArt-$\Sigma$ and Hunyuan are more challenging problems than PixArt-α. ViDiT-Q [1] shows that naive quantization only slightly decreases PixArt-α’s performance, while severely decreasing PixArt-$\Sigma$’s performance.
>
> >**W3. In ablation studies, the paper only studies hyperparameter lambda. In addition to that, the paper should ablate the components of the proposed method: attention rescaling and residual caching.**
>
> 1. Regarding attention rescaling and residual caching, these are not independent modules but are jointly designed as the core of our method. The attention rescaling recovers the attention distribution within the sliding window, but deviates from the intended probability formulation (sum probability < 1). Hence, we propose to apply the cached residual from the masked tokens, thereby compensating for the missing probability.
> 2. We ablate the components of Re-ttention, and evaluate them on CogVideoX T2V models. All sparsity is set to 96.9% to be consistent with the Table 1 result. Removing either residual caching or attention scaling degrades all the evaluation metrics severely, re-affirming that Re-ttention is a holistic new method based on different mathematical assumptions than DiTFastAttn. In Re-ttention, all the components are indispensable and need to work together to achieve good performance.
>
> | Attention     | PSNR ↑ | SSIM ↑ | LPIPS ↓ | ImageQual ↑ | SubConsist ↑ |
> |---|---|---|---|---|---|
> | Re-ttention   | 27.96  | 0.894  | 0.059    | 64.87%       | 94.80%         |
> | No Residual   | 14.51  | 0.563  | 0.602    | 37.89%       | 94.49%         |
> | No Scaling    | 2.64   | 0.443  | 0.606    | 39.50%       | 90.02%         |
>
> >**Q1. Is the degradation of DiTFastAttn on PixArt-Sigma related to token compression? How does Re-ttention handle this change?**
>
> We use the pipeline in HuggingFace’s diffusers to run our experiments. And to our knowledge, this token compression technique has not been implemented.
>
> [1] T. Zhao, T. Fang, H. Huang, et al., “Vidit-Q: Efficient and accurate quantization of diffusion transformers for image and video generation,” arXiv preprint arXiv:2406.02540, 2024.

---

> > ### Comment · Reviewer_SARk · 2025-08-04
> > **follow-up**
> >
> > The author explains that DiTFastAttn failed on PixArt-Sigma and Hunyuan with > 90% sparsity. If the results at 75% sparsity were included in the experiment, it would help rule out other factors.
> >
> > What's the trade-off between quality and speed at different sparsities? This will help understand the significance of pushing sparsity down to 95%.

---

> ### Author Response · Authors · 2025-08-05
>
> Thanks to the reviewer for the thoughtful suggestions. Here we add DiTFastAttn and Re-ttention’s T2I GenEval results for PixArt-$\Sigma$ and Hunyuan at 75% sparsity and 87.5% sparsity. We can see DiTFastAttn's performance degrades dramatically and fails to generate quality images when sparsity is greater than 87.5%. For example, a GenEval score of 0.411 (achieved by DiTFastAttn at 93.8% sparsity on PixArt-$\Sigma$) corresponds to the generated image in Figure 6 (row 2 col 4) in the manuscript, which has unacceptable quality. Similarly, a GenEval score of 0.024 (achieved by DiTFastAttn at 93.8% sparsity on Hunyuan) corresponds to the generated image in Figure 6 (row 3 col 4), which fails completely. However, Re-ttention can still maintain good generation quality at >95% sparsity; examples can be found in Figure 6 (row 2, 3 col 6, 7). This shows a clear advantage of Re-ttention over DiTFastAttn at >=87.5% sparsity and the significance of our proposed techniques to preserve quality for ultra sparse attention.
>
> **PixArt-$\Sigma$**
>
> | Method       | 75%   | 87.5% | 93.8% | 96.9% |
> |--------------|-------|-------|-------|-------|
> | DiTFastAttn  | 0.545 | 0.517 | 0.411 | 0.233 |
> | Re-ttention  | 0.547 | 0.547 | 0.513 | 0.512 |
>
> **Hunyuan**
>
> | Method       | 75%   | 87.5% | 93.8% | 96.9% |
> |--------------|-------|-------|-------|-------|
> | DiTFastAttn  | 0.590 | 0.453 | 0.024 | 0.002 |
> | Re-ttention  | 0.593 | 0.594 | 0.585 | 0.590 |
>
> We further augment Table 4 by adding the latency (ms) of Re-ttention (one attention block) at 75%, 87.5% and 93.8% sparsity. We can see 96.9% sparsity (achievable by Re-ttention) has a more than 2x speedup over 87.5% sparsity (the sparsity that DiTFastAttn operates on). Thus, there is a significant motivation and efficiency benefit to further push sparsity to >95% while maintaining the quality.
>
> | Model        | Full-Attn | 75%   | 87.5% | 93.8% | 96.9% |
> |--------------|-----------|-------|-------|-------|-------|
> | PixArt | 2.59      | 0.752 | 0.421 | 0.268 | 0.186 |
> | Hunyuan      | 2.61      | 0.762 | 0.444 | 0.287 | 0.206 |
> | CogVideoX    | 15.92     | 4.779 | 2.730 | 1.840 | 1.277 |
>
> We will add these results in the revised manuscript, and further clarify the visual examples provided throughout the paper, which empirically show the significant performance advantage of our method over existing methods.

---

> > ### Comment · Reviewer_SARk · 2025-08-07
> >
> > Thanks for providing the additional comparisons to support the paper. I would consider raise my score.

---

### Official Review · Reviewer_Pwod · 2025-07-03

**Clarity:** 2
**Significance:** 3
**Originality:** 3
**Rating:** 4
**Confidence:** 4

**Summary:**

This paper proposes Re-ttention, a method to accelerate pre-trained diffusion transformers based on ultra sparse attention computation. Previous methods like DiTFastAttn typically fail in the scenarios of highly sparse attention patterns. This method finds the source of such failures is on the denominator of softmax and takes this factor into consideration to address the problem. Experiments demonstrate its robustness and superiority over state-of-the-art baselines.

**Questions:**

* The acceleration is compared against naive full attention implementation or fused attention in PyTorch 2.0 or higher?

And please refer to the weaknesses part above.

**Ethical Concerns:**

["NO or VERY MINOR ethics concerns only"]

**Final Justification:**

The paper presents some valid exploration in image generation with sparse attention. The authors' response clear most of my concerns. I lean towards the positive side.

**Limitations:**

The authors have discussed the limitations in the supplement.

**Paper Formatting Concerns:**

I did not find issues in this aspect.

**Quality:**

3

**Strengths And Weaknesses:**

### Strengths:

1. The method is well motivated and novel. The authors provide sufficient analysis to support the source of the focused issue, which is on the denominator of softmax. Successfully, after taking it into consideration, the problem is largely alleviated according to the experiments.
2. The experiments are extensive. Almost all popular DiT-based models can be accelerated by the proposed method.
3. The codes are attached as supplementary materials. I assume that they will be released if the paper is accepted.

### Weaknesses:

1. It seems that the method is a patched version of DiTFastAttn. Although effective, it does not introduced too much difference. For example, it still largely relies on attention residual proposed in DiTFastAttn.
2. The logical flow can be improved. It would be better to illustrate what has been achieved by previous methods, like identifying the important regions and using cached residual, then analyze what is the drawbacks of them, and finally present the proposed method.
3. The implementation has a large room of improvement. I have a glance of the codes. It seems that the authors adopt naive implementation to extract the attention score and compute the window attention, without using fused kernels like FlashAttention. I wonder if this will consume more GPU memory. And is there any solution to further enhance the efficiency using fused kernels or the api of flexattention in PyTorch?

---

> ### Author Rebuttal · Authors · 2025-07-31
>
> >**W1. It seems that the method is a patched version of DiTFastAttn. Although effective, it does not introduced too much difference. For example, it still largely relies on attention residual proposed in DiTFastAttn.**
>
> Our method is not simply a patched version of DiTFastAttn, but rather addresses an important limiting factor in existing sparse attention schemes and has not appeared in any of the sparse attention acceleration techniques that the literature had considered. First, DiTFastAttn proposes to use a combination of attention acceleration techniques, including attention residual caching, attention sharing across timesteps, and CFG, etc. Their work focuses on deciding which technique should be used for each layer of DiT and for example, achieves only 87.5% sparsity (using ⅛ sequence length as window size in their paper) without FID degradation after a greedy search. Here we re-evaluate their method on GenEval T2I benchmark in Figure 3, which confirms that DiTFastAttn only achieves up to 87.5% sparsity without degrading generation quality.
>
> In this paper, we point out that a key limitation of the existing sparse attention schemes is that the denominator of Softmax is decreased due to applying sparse attention, which “causes the attention score distribution to shift away from what would be expected in full attention” [1]. Motivated by this, our Re-ttention proposes statistical attention reshaping which aims to recover the original Softmax denominator before sparsity is applied by estimating and calibrating the denominator from step to step (see Sec. 4.2) through a cached $\rho$ value plus ramp-up hyperparameter $\lambda$. As a result, we achieve as high as 96.875% sparsity without degrading the T2I quality on GenEval (Figure 3), largely outperforming DiTFastAttn.
>
> Furthermore, the cached attention residual in our paper is NOT the same as that in DiTFastAttn. There is a key difference. While existing methods like DiTFastAttn define the residual as the difference (subtraction) between full attention and sparse attention, our Re-ttention defines it as the attention contribution of masked tokens only (those outside the sparse attention window)—which is a critical distinction and completely different from DiTFastAttn and other previous residual caching techniques.
>
> For example, consider a row of attention scores: $\text{softmax}(QK) = [0.1, 0.4, 0.4, 0.1]$ in full attention. With residual caching for sparse attention in DiTFastAttn, we have the row of sparse attention scores: $\text{softmax}(QK) = [0, 0.5, 0.5, 0]$, and residual $R =  [0.1, 0.4, 0.4, 0.1] - [0, 0.5, 0.5, 0] = [0.1, -0.1, -0.1, 0.1]$ is cached to the next step. However, in Re-ttention, the row of $\text{softmax}(QK) = [0, 0.4, 0.4, 0]$ after rescaling the denominator with our statistical reshaping, and the cached residual is $R’ = [0.1, 0, 0, 0.1]$, which contains only the attention score of masked tokens outside of the sparsity window. The former caching scheme in existing literature relies on the assumption that R changes minimally over steps, which however is not necessarily true within the sparse attention window and is prone to contaminating the attention scores within the window if in the next step, e.g., $[0.1, 0.4, 0.4, 0.1]$ changes to $[0.1, 0.3, 0.5, 0.1]$. In contrast, our residual is mathematically equivalent to caching only the attention scores outside the sparse window, and in the meantime focuses on updating only the important attention scores within the window and approximating the full attention via statistical reshaping (through cached $\rho$). A detailed explanation can be found in Appendix A.3.
>
> In other words, Re-ttention proposes a holistic new method with two parts: 1) approximating the denominator via a reused $\rho$ value to rescale the attention scores within the attention window, 2) caching the true residuals outside the window to represent the residual attention of masked tokens. Both parts are necessary and work together to yield a much closer approximation to original full attention, without contaminating attention values due to distribution shift.
>
> Finally, experimental results show that Re-ttention can achieve up to 96.9% sparsity with non-degraded generation quality and substantially outperforms DiTFastAttn in visual quality, demonstrated by the extensive T2I and T2V examples provided in Figures 1, 6, 9, 10, 11, 12, 13, 14, 15. In fact, more examples confirm the same observation, which are not listed due to space limit. This again confirms that there is a fundamental limitation of the caching scheme adopted by DiTFastAttn, which does not uphold its assumption, and the significance of our Re-ttention technique.
>
> >**W2. The logical flow can be improved. It would be better to illustrate what has been achieved by previous methods, like identifying the important regions and using cached residual, then analyze what is the drawbacks of them, and finally present the proposed method.**
>
> We thank the reviewer for their suggestions. We will definitely aim to revise and better illustrate the approach of Re-ttention and existing methods, and then provide contrast of the contributions, such as the statistical reshaping method and the minimal performance degradation observed under ultra-sparse attention.
>
> >**W3. The implementation has a large room of improvement. I have a glance of the codes. It seems that the authors adopt naive implementation to extract the attention score and compute the window attention, without using fused kernels like FlashAttention. I wonder if this will consume more GPU memory. And is there any solution to further enhance the efficiency using fused kernels or the api of flexattention in PyTorch?**
>
> The implementation we uploaded uses the original Pytorch implementation instead of specialized fused kernels, this work focuses on showing and verifying that the proposed statistical reshaping mechanism is critical and necessary to maintain generation quality for DiT under ultra-sparse attention on a wide range of T2I and T2V benchmarks, and its performance comparison to existing methods, rather than on kernel development.
>
> Note that as our method introduces a post-Softmax attention rescaling step, it is not compatible with existing kernel solutions like FlexAttention, which only provides flexibility before the Softmax operation. That said, we believe that Re-ttention can be adapted to incorporate kernel fusion with further non-trivial engineering efforts, which is an interesting direction for future work.
>
> >**Q1. The acceleration is compared against naive full attention implementation or fused attention in PyTorch 2.0 or higher?**
>
> They are compared against the fused attention in PyTorch 2.0.
>
> [1] G. Xiao, Y. Tian, B. Chen, S. Han, and M. Lewis, “Efficient streaming language models with attention sinks,” arXiv preprint arXiv:2309.17453, 2023.

---

> > ### Comment · Reviewer_Pwod · 2025-08-07
> >
> > Thanks for the response. It addresses most of the concerns. I remain positive to the submssion.

---

> ### Author Response · Authors · 2025-08-06
>
> Dear Reviewer Pwod,
>
> We are wondering if our rebuttal has addressed your main concerns. We are committed to answering any further questions you may have and clarifying the manuscript before the rebuttal deadline.

---

### Official Review · Reviewer_wos9 · 2025-07-03

**Clarity:** 3
**Significance:** 3
**Originality:** 3
**Rating:** 4
**Confidence:** 4

**Summary:**

This paper introduces Re-ttention, a sparse attention mechanism that addresses the computational inefficiencies of DiT for visual generation. By leveraging the temporal redundancy in Diffusion Models, Re-ttention reshapes attention scores using prior softmax distribution history, achieving extremely high sparsity while preserving the visual quality. It outperforms state-of-the-art methods like FastDiTAttn and Sparse VideoGen in both quality and efficiency. Experimental results on T2V/T2I models such as CogVideoX and PixArt DiTs validate its effectiveness, offering a significant step forward in sparse attention for visual generation.

**Questions:**

see above

**Ethical Concerns:**

["NO or VERY MINOR ethics concerns only"]

**Final Justification:**

The rebuttal has addressed most of my concerns. I will maintain my borderline accept rating.

**Limitations:**

see above

**Quality:**

3

**Strengths And Weaknesses:**

**Strengths**
1. The paper is well-motivated, with abundant analyses and examples that clearly demonstrate the reasoning behind its technical designs.
2. It provides extensive experimental results across various models, effectively showcasing its robustness and effectiveness.
3. The figures are highly informative and significantly aid in understanding the core concepts of the paper.

**Weaknesses**
1. The paper introduces a hyperparameter, \lambda. While the ablation study indicates that a non-zero value for \lambda is beneficial, the performance difference between \lambda = 0 and the optimal \lambda = 0.04 is not substantial. A more thorough explanation of the rationale behind this parameter would enhance the paper.
2. Re-ttention relies on periodically performing full attention computations to refresh its cache. The paper follows a fixed schedule. However, a more in-depth analysis of the trade-offs here would be beneficial.

---

> ### Author Rebuttal · Authors · 2025-07-31
>
> > **W1. The paper introduces a hyperparameter, $\lambda$. While the ablation study indicates that a non-zero value for $\lambda$ is beneficial, the performance difference between $\lambda$ = 0 and the optimal $\lambda$ = 0.04 is not substantial. A more thorough explanation of the rationale behind this parameter would enhance the paper.**
>
> The rationale behind this parameter is that DiT tends to focus on global structures during the early denoising steps and shifts its attention to finer local details in the later steps. In other words, the attention in DiT progressively shifts from long-range tokens to local tokens over the denoising steps, thereby increasing the denominator ratio $\rho$. It also matches our observation (see Figure 5) about the ratio $\rho$. Therefore, we introduce a ramp-up hyperparameter $\lambda$ to provide a better approximation of the parameter.
>
> > **W2. Re-ttention relies on periodically performing full attention computations to refresh its cache. The paper follows a fixed schedule. However, a more in-depth analysis of the trade-offs here would be beneficial.**
>
> Here we add the result on PixArt-$\Sigma$ with 96.9% sparsity after changing the sampling schedule $T$ from 5 to 3 and 7. A smaller $T$ means more inferences in full attention and more accurate cached residual, thereby resulting in better performance. Our Re-ttention outperforms DiTFastAttn, regardless of $T$.  We choose $T$=5 to balance the trade-off between computational efficiency and generation performance, while maintaining image/video quality, making it a practical and generalizable choice across different models and tasks.
>
> | Attention     | T = 3 GenEval ↑ | T = 3 HPSv2 ↑ | T = 5 GenEval ↑ | T = 5 HPSv2 ↑ | T = 7 GenEval ↑ | T = 7 HPSv2 ↑ |
> |---------------|------------------|----------------|------------------|----------------|------------------|----------------|
> | DiTFastAttn   | 0.376            | 26.81          | 0.233            | 22.79          | 0.147            | 20.87          |
> | Re-ttention   | 0.544            | 29.26          | 0.512            | 27.72          | 0.479            | 26.16          |

---

> ### Author Response · Authors · 2025-08-06
>
> Dear Reviewer wos9,
>
> We are wondering if our rebuttal has addressed your main concerns. We are committed to answering any further questions you may have and clarifying the manuscript before the rebuttal deadline.

---

> > ### Comment · Reviewer_wos9 · 2025-08-06
> > **Response**
> >
> > The rebuttal has addressed most of my concerns. I will maintain my borderline accept rating.

---

### Note · Authors · 2025-08-16

We thank the reviewers for their constructive comments and thoughtful discussions. We are glad that the contributions of our proposed method, Re-ttention, is recognized by the following notes from the reviewers:

- It is well-motivated (wos9) and novel (Pwod) to perform statistical attention reshaping for ultra sparse attention in DiT.
- This paper provides a sufficient analysis (wos9, Pwod) of the reason behind reshaping attention scores.
- This paper conducts extensive experiments (wos9, Pwod, phiV) to showcase the robustness and effectiveness of Re-ttention.
- The method shows better results across multiple metrics (SARk) as compared to other baseline methods.
- The method achieves a high sparsity of 96.9% with minimal quality loss (phiV).

During the rebuttal, we tried to meticulously address the reviewers’ concerns by clarifying the extent of certain limitations and providing additional experimental results. Specifically, we would highlight the following responses:

- Regarding novelty (SARk, Pwod): Through detailed explanation and examples with intuitions, we have clarified the novelty of Re-ttention and its significant difference from existing methods. We conducted more ablation studies to demonstrate that Re-ttention is a holistic new method with all its components necessary and effective, rather than extending existing work. The uniqueness and strengths of Re-ttention are supported by the extensive experimental results.
- More ablation study (wos9, SARk, phiV): We have extended our ablation study to include additional experiments that examine the impact of the sampling schedule, individual components in our method, and different mask selection strategies.
- Clarifying baseline results (SARk): We have analyzed why different DiT models have different degrees of degradation under some baseline method, and also provided additional experiments to support it.
- Theoretical intuition (phiV): We have provided an explanation with mathematical intuitions, to clarify how Re-ttention offers a more accurate approximation of full attention than existing sparse attention and residual caching methods for DiT.

We believe that the discussions have helped strengthen the scientific rigor of this work and enhance the presentation of our contribution. This insightful exchange has helped us to brainstorm potential future research directions which we look forward to exploring. We thank the reviewers for their valuable feedback and support!

Best Regards,

Authors

---

### Decision · Program_Chairs · 2025-09-17

**Decision:**

Accept (poster)

**Comment:**

This paper introduces Re-ttention, a sparse attention mechanism that achieves ultra-high sparsity (up to 96.9%) with minimal quality degradation in diffusion transformers. The method is well-motivated, proposing a statistically grounded correction of softmax denominator shift. The authors provide extensive experiments across T2I and T2V models and strong ablations, showcasing both effectiveness and generality. During the rebuttal, the authors addressed most of reviewers’ concerns, including theoretical intuitions, implementation details, comparisons, and memory/efficiency trade-offs. Reviewers converged on borderline-to-positive recommendations. With strong empirical results and its potential for broad applicability, I recommend acceptance.